# Lysosomal protein transmembrane 5 promotes lung-specific metastasis by regulating BMPR1A lysosomal degradation

Bo Jiang[1,5], Xiaozhi Zhao[1,5], Wei Chen[1], Wenli Diao[1], Meng Ding[1], Haixiang Qin[1], Binghua Li[2], Wenmin Cao[1], Wei Chen[1], Yao Fu[3], Kuiqiang He[1], Jie Gao[1], Mengxia Chen[1], Tingsheng Lin[1], Yongming Deng[1], Chao Yan[4✉] & Hongqian Guo[1✉]

Organotropism during cancer metastasis occurs frequently but the underlying mechanism remains poorly understood. Here, we show that lysosomal protein transmembrane 5 (LAPTM5) promotes lung-specific metastasis in renal cancer. LAPTM5 sustains self-renewal and cancer stem cell-like traits of renal cancer cells by blocking the function of lung-derived bone morphogenetic proteins (BMPs). Mechanistic investigations showed that LAPTM5 recruits WWP2, which binds to the BMP receptor BMPR1A and mediates its lysosomal sorting, ubiquitination and ultimate degradation. BMPR1A expression was restored by the lysosomal inhibitor chloroquine. LAPTM5 expression could also serve as an independent predictor of lung metastasis in renal cancer. Lastly, elevation of LAPTM5 expression in lung metastases is a common phenomenon in multiple cancer types. Our results reveal a molecular mechanism underlying lung-specific metastasis and identify LAPTM5 as a potential therapeutic target for cancers with lung metastasis.

[1] Department of Urology, Nanjing Drum Tower Hospital, the Affiliated Hospital of Nanjing University Medical School, Institute of Urology, Nanjing University, Nanjing, Jiangsu 210008, China. [2] Department of Hepatobiliary Surgery, Nanjing Drum Tower Hospital, the Affiliated Hospital of Nanjing University Medical School, Nanjing, Jiangsu 210008, China. [3] Department of Pathology, Nanjing Drum Tower Hospital, the Affiliated Hospital of Nanjing University Medical School, Nanjing, Jiangsu 210008, China. [4] State Key Laboratory of Pharmaceutical Biotechnology, School of Life Sciences, Nanjing University, Nanjing, China. [5]These authors contributed equally: Bo Jiang, Xiaozhi Zhao. ✉email: yanchao@nju.edu.cn; dr.ghq@nju.edu.cn

Metastasis is involved in 90% of cancer-related mortality and blocking metastasis remains a major challenge for cancer treatment[1]. In clinical practice, it is frequently observed that primary tumors have the proclivity to often metastasize and colonize specific organs, for instance, breast cancer (BCa) often metastasizes to the brain, prostate cancer (PCa) to the bone and renal cancer (RCC) to the lung[2,3]. Tumors with different organ metastases often show distinct responses to treatments and prognoses[4,5], the mechanism of which is not well understood, presenting an unanswered scientific question and a promising therapeutic opportunity for cancer treatment.

Metastatic dissemination occurs frequently in the early stage of tumor development but the clinical manifestation of metastases often takes years[6–8] because a vast majority of tumor cells that infiltrate the parenchyma of an organ are rejected by organ-specific inhibitory signals (e.g., bone morphogenetic proteins (BMPs) in the lung stroma; Wnt family member 5a (Wnt5a), transforming growth factor Beta 2 (TGF-β2) and growth differentiation factor 10 (GDF10) in the bone stroma), or aboriginal cells (like astrocytes in the brain parenchyma and osteoblasts in the bone stroma), and are eventually destroyed or enter a dormant state[9–12]. However, certain genetic mutations or molecular characteristics enable a small fraction of tumor cells, sometimes called metastasis-initiating cells, to interact with aboriginal cells and/or overcome these inhibitory signals, and maintain cancer stem cell-like traits and eventually colonize target organs[13–17]. It is now known that unique expression patterns of certain molecules in cancer cells mediate their selective interaction with target organs and drive organ-specific metastasis. For example, Coco, a BMP antagonist, mediates lung-specific metastasis, and Cx43-PCDH7 junction formation triggers brain metastasis in BCa[15,18–21]. However, the molecular basis underlying lung-specific metastasis in most cancers remains unclear. In RCC, for instance, metastases from different organs display unique genetic alterations. Loss of chromosome 8p and 18q, along with gain of chromosome 12, are found very frequently in lung metastases[22,23]. In particular, organ-specific metastases not only share the vast majority of mutations with, but also possess characteristic evolutionary offshoots of, their parental primary tumors and other organ metastases[24–26]. Moreover, patients with lung metastasis seem to show a better survival rate, whereas bone and liver metastasis are associated with worse tyrosine kinase inhibitor (TKI) response and poor prognosis[27–29]. The cause of this clinical manifestation is still unclear.

In this work, we explore the mechanism that drives the lung-specific metastasis of RCC. We find that lysosomal protein transmembrane 5 (LAPTM5) promotes lung-specific metastasis in RCC patients. Mechanistic studies suggest that LAPTM5 exerts this function by promoting lysosomal sorting and degradation of BMP receptor 1A (BMPR1A), thereby enhancing the self-renewal capability of metastasis-initiating cells in the lung microenvironment.

## Results

### Establishment of RCC cell lines with high lung metastasis tendency.
We first generated organ-specific metastatic derivatives of RCC by repeated intracardiac (I.C.) inoculation and metastases clone selection/expansion, using a luciferase-labeled murine RCC cell line Renca[luci] (Fig. 1a). In keeping with the clinical manifestations of RCC[2], the Renca[luci] cells mainly metastasized to the lung, bone, and brain (Supplementary Fig. 1a, b and Fig. 1b). The resulting subpopulations were named by their source organ and generation, e.g., lung metastatic derivative-1a, Renca[LuM1a], is a lung metastatic derivative after the first round of selection (Fig. 1c). Intriguingly, different organ derivatives showed distinct cell morphology but similar proliferation rates in vitro

(Supplementary Fig. 1c, d). The above experiments were also repeated with the human RCC cell line 786O[luci/eGFP] (dual-labeled with luciferase and eGFP) in the immunodeficient NOD/SCID mice but only the lung derivatives were selected (Supplementary Fig. 1e, f), similar results to that of the murine cell line were observed (Supplementary Fig. 1g, h).

To assess the lung metastasis-forming ability of the organ-specific derivatives, representative Renca[luci] parental and derivative cells were inoculated into the arterial circulation of BALB/c mice and analyzed for lung metastasis. The lung-met derivative Renca[LuM2b] metastasized to the lung in 75% (6/8) of the mice 11 days post injection, compared to 10% (1/10) in the parental line, 12.5% (1/8) in the brain-met Renca[BrM2b], and 0% (0/9) in the bone-met Renca[BoM2] cell line (Fig. 1d). Mice inoculated with Renca[LuM2b] cells exhibited shorter survival than the parental Renca[luci] group (Fig. 1e). Increased lung metastasis activity was also confirmed by ex vivo imaging of excised lungs in mice inoculated with the Renca[LuM2b] or 786O[LuM1a] derivatives (Fig. 1f, g). These data demonstrated that we have successfully modeled the multi-organ metastasis of RCC and obtained RCC cell derivatives with high lung metastasis tendency.

### LAPTM5 mediates lung metastasis, but not brain or bone metastasis.
We next performed genome-wide transcriptional profiling to identify molecules associated with different metastatic phenotypes. Principal component analysis (PCA) revealed the adjacent clustering of the Renca[LuM2b] derivative with the Renca[BrM2b] derivative, away from the Renca[luci] cells and the Renca[BoM2] derivative (Supplementary Fig. 2a, b and Fig. 2a); this is consistent with a previous study, which suggested the presence of common mediators in pulmonary and cerebral metastases[20]. To identify the key molecules associated with lung-specific metastasis, we compared the Renca[LuM2b] derivative with Renca[luci], Renca[BoM2] and Renca[BrM2b], and identified 562, 942 and 276 genes, respectively, that were highly expressed in Renca[LuM2b] (Supplementary Fig. 2c–e). Of these upregulated genes, 69 were present in all sets (Fig. 2b and Supplementary Data 1). In addition, we analyzed an independent human RCC dataset (Jon_Renal_Cancer)[30] and found 760 probes/genes that displayed higher expression levels in lung metastases (L-Mets) compared with primary RCC (pri-RCC) (Supplementary Fig. 2f, g). Integrated analysis of these two datasets identified three genes: cathepsin S (CTSS), lysosomal protein transmembrane 5 (LAPTM5), and insulin-like growth factor binding protein 5 (IGFBP5), that appeared to be activated specifically in RCC cells from lung metastases in both datasets (Fig. 2b and Supplementary Data 2). Among these three genes, LAPTM5 showed much more prominent elevation than CTSS and IGFBP5 in lung derivatives (Fig. 2c). Moreover, LAPTM5, a lysosomal-associated multi-spanning membrane protein[31], not only exhibited higher transcription levels in the Renca[LuM2b] lung derivative, but also in clinical specimens from the L-Mets and primary RCC with lung metastasis (RCCL) subgroups, compared with the pri-RCC subgroup (Supplementary Data 2). Importantly, elevated transcription and translation of the LAPTM5 gene was only observed in the lung metastasis derivatives but not in brain or bone metastasis (Fig. 2c, d). These data suggested that LAPTM5 might be associated with lung metastasis of RCC. These results were also validated by immunohistochemical (IHC) analysis of different tissues in the mouse model (Fig. 2e). To strengthen these findings, we directly injected Renca[luci] cells through mice tail vein and isolated three generations of cell derivatives from lung metastases; higher LAPTM5 levels were detected in two of three metastatic clones, further confirming the differential expression of LAPTM5 in lung metastases (Supplementary Fig. 3a, b).

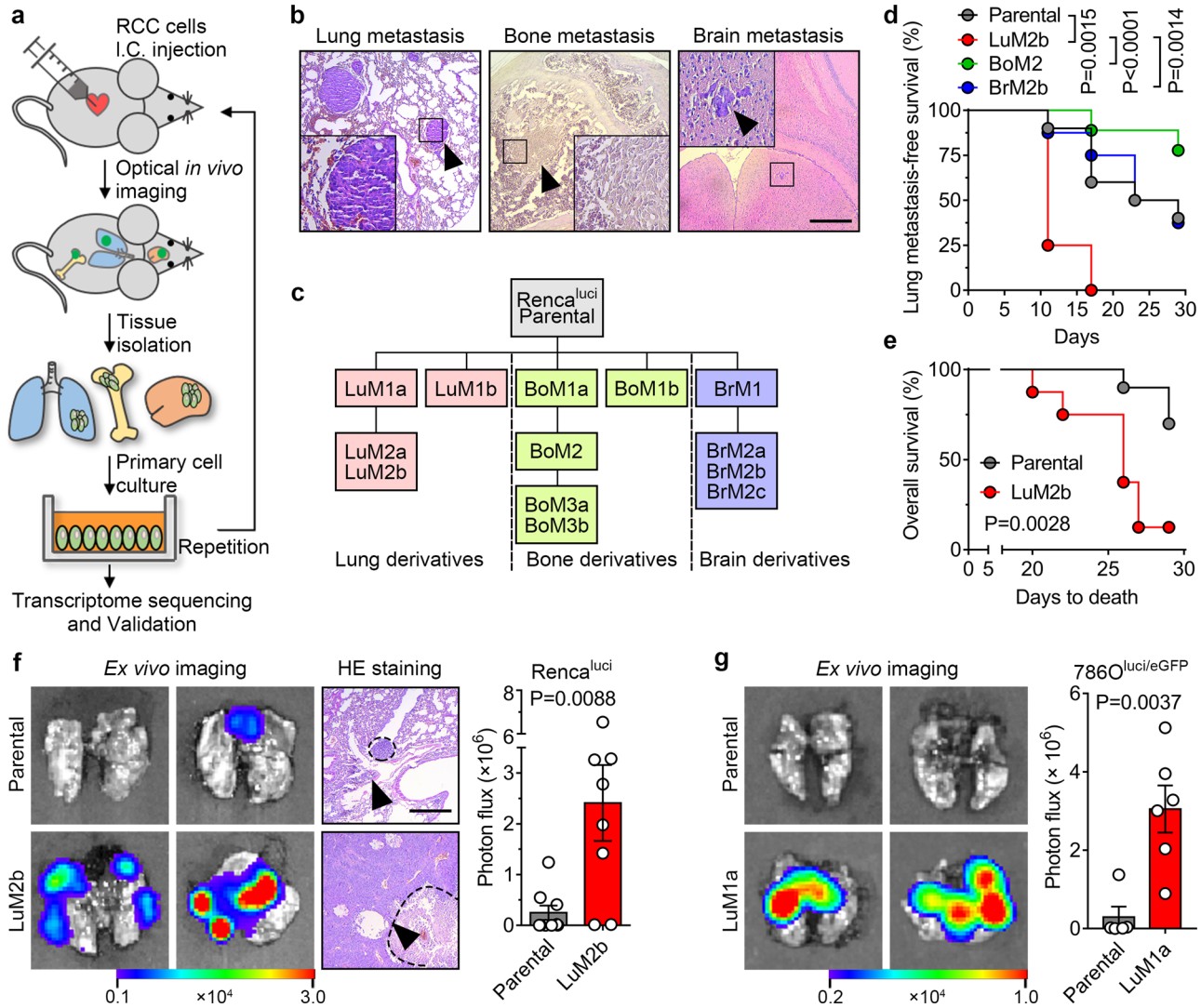

**Fig. 1 Isolation, characterization, and analysis of lung metastatic derivatives. a** Construction of the multi-organ metastasis model and screening of organospecific metastatic cell derivatives. I.C., intracardiac. **b** Hematein-eosin (H&E) staining of lung, bone, and brain metastases (arrowheads) by Renca[luci] cells. Scale bar, 400 μm. **c** Flowchart of the in vivo selection of lung, bone, and brain-specific metastatic subpopulations from Renca[luci] cells. **d** Kaplan–Meier survival curves for lung metastasis-free survival of Renca[luci] cells and representative derivatives (Parental, $n = 10$ mice; LuM2b, $n = 8$ mice; BoM2, $n = 9$ mice; BrM2b, $n = 8$ mice). **e** Kaplan–Meier survival curves for overall survival of Renca[Parental] and Renca[LuM2b] cells (Parental, $n = 10$ mice; LuM2b, $n = 8$ mice). **f** Representative ex vivo bioluminescent and H&E images of lung metastases at 30 days after I.C. injection of Renca[luci] cells (Parental, $n = 9$ mice; LuM2b, $n = 8$ mice; left panel) and quantification of the photon flux (right panel). Scale bar, 400 μm. **g** Representative ex vivo bioluminescent images of lung metastases at 90 days after I.C. injection of 786O[luci/eGFP] cells (Parental, $n = 5$ mice; LuM1a, $n = 6$ mice; left panel) and quantification of the photon flux (right panel). In **f** and **g**, the data are presented as mean ± SEM. Two-sided log-rank test was used for statistical analysis of (**d**) and (**e**), two-tailed Student's unpaired $t$-test for (**f**) and (**g**). Source data are provided as a Source data file.

To determine whether LAPTM5 contributes to organotropic metastasis, we stably overexpressed *Laptm5* in Renca[luci] cells and knockdown *Laptm5* in Renca[LuM2b] cells, respectively (Fig. 2f), and tested their metastasis potential in vivo (Fig. 2g). Overexpression of *Laptm5* in Renca[luci] cells markedly accelerated lung metastasis (Fig. 2h, k), whereas knockdown of *Laptm5* in Renca[LuM2b] cells significantly suppressed lung metastasis (Fig. 2l, o). In contrast, the bone and brain metastasis ability of both Renca[luci] and Renca[LuM2b] cells were not affected by *Laptm5* expression levels (Fig. 2i, j, m, n, p). A parallel experiment carried out in the human RCC cell lines 786O[Luci/eGFP] and 786O[LuM1a]/786O[LuM1b] also revealed similar roles of LAPTM5 in mouse models of lung metastasis (Supplementary Fig. 3c–f). Moreover, ectopically expressed LAPTM5 in Renca[luci] bone- and brain-derivatives converted their organotropism and enhanced their ability to metastasize to the lung (Supplementary Fig. 3g, h). Together, these results established that LAPTM5 is a positive regulator of renal cancer lung metastasis.

**LAPTM5 promotes self-renewal and cancer stem cell-like traits of RCC cells**. Proliferation, epithelial-mesenchymal transition (EMT) plasticity, resistance to apoptosis, and self-renewal in foreign microenvironment are vital intrinsic properties of metastasizing tumor cells[32,33]. Therefore, we investigated the effect of LAPTM5 on each of these cellular processes. Firstly, altering the expression of LAPTM5 in RCC cells did not affect the in vitro proliferation rate (Supplementary Fig. 4a–c). Secondly, LAPTM5 neither induced the EMT process nor enhanced the invasiveness of Renca cells or 786-O cells (Supplementary Fig. 4d, e). Thirdly, no notable apoptosis-resistance effect of LAPTM5 in Renca[luci] cells was observed in vivo (Supplementary Fig. 4f). Lastly, to test

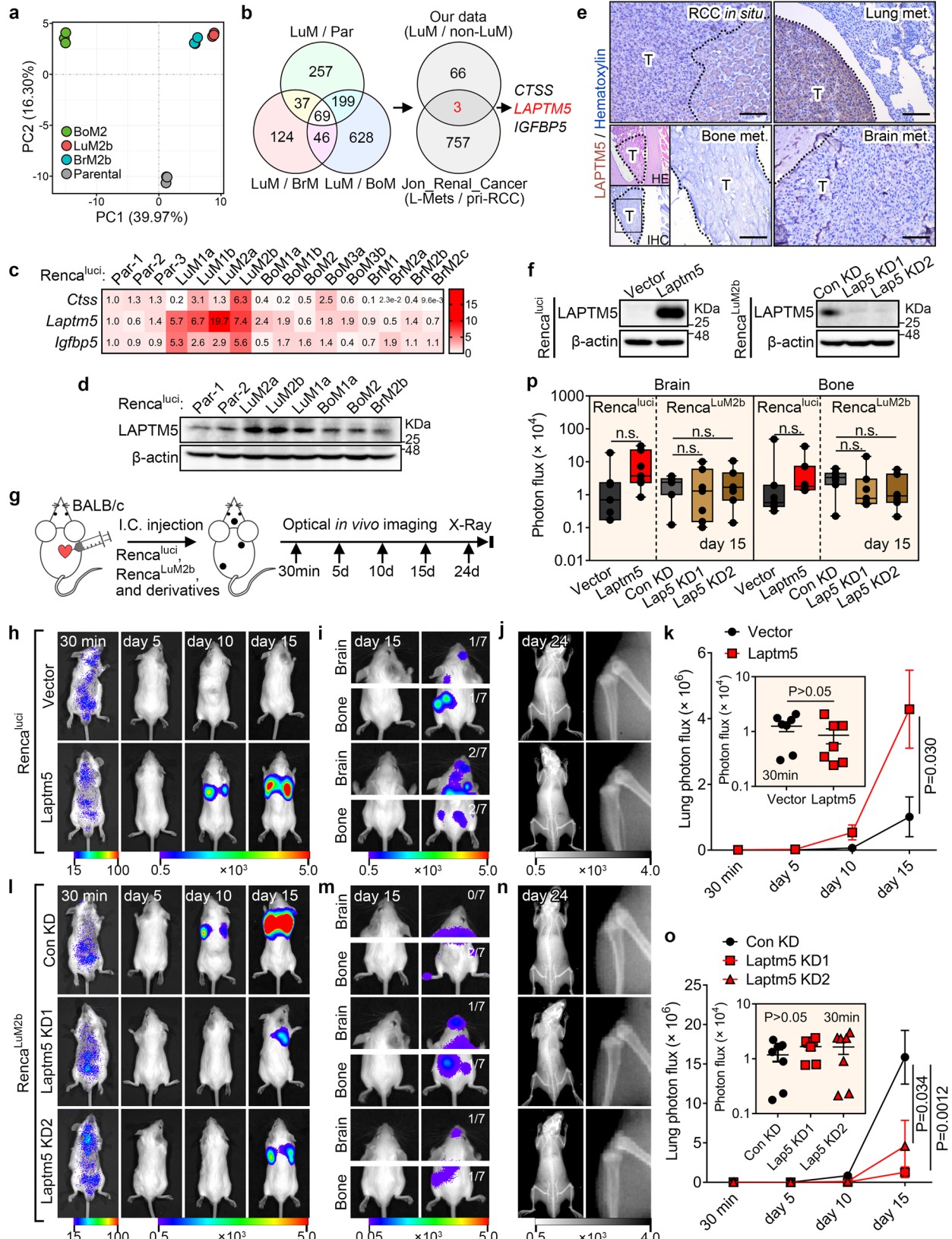

whether LAPTM5 is associated with self-renewal ability of cancer cells, we established the forced lung metastasis model by directly injecting Renca[luci] and derivative cell lines into mouse tail vein and collected the lungs at different time points (Fig. 3a). IHC staining of luciferase in lung sections revealed that tumor cells extravasated in the stroma of the lung as early as one day after injection (Fig. 3b).

LAPTM5 overexpression did not enhance the ability of Renca[luci] cells to infiltrate the lung (Supplementary Fig. 4g), but accelerated the formation of both micro-metastases (maximum diameter < 100 μm) and macro-metastases (maximum diameter > 100 μm) since day 7 (Fig. 3b–d). On the contrary, silencing of *Laptm5* suppressed the metastatic outgrowth of Renca[LuM2b] cells in the lung stroma

**Fig. 2 LAPTM5 mediates lung-specific metastasis. a** Principal component analysis (PCA) of Renca[luci] parental and derivative cells. **b** Overlap of up-regulated genes in Renca[LuM2b] cells (LuM) as compared to Renca[Parental] (Par), Renca[BrM2b] (BrM) and Renca[BoM2] (BoM) cells (left panel) and further overlap with up-regulated genes in RCC lung metastases (L-Mets) as compared with primary RCC without lung metastases (pri-RCC) in the Jon_Renal_Cancer dataset (right panel). **c** Heatmap showing the mRNA expression of *Ctss*, *Laptm5* and *Igfbp5* in Renca[luci] parental and derivative cells measured by qRT-PCR, relative gene expression in each cell was compared with parental-1 (Par-1). **d** Immunoblotting (IB) analyses of LAPTM5 expression in Renca[luci] parental and derivative cells. **e** Immunohistochemistry (IHC) analysis of LAPTM5 staining in RCC in situ (renal subcapsular injection) and lung, bone, brain metastases. HE staining was used to show the site of bone metastasis. T, tumor. Scale bar, 200 μm. **f** IB analysis of control and Laptm5-overexpressing Renca[luci] cells or control and Laptm5-silenced Renca[LuM2b] cells. **g** Schematic illustration for intracardiac cell inoculation and metastasis detection. Representative bioluminescent images of lung metastases (**h**), brain and bone metastases (**i**) in mice treated as in (**g**) with Renca[luci] and derivatives. **j** Representative X-ray images of bone metastases in mice treated as in (**g**). **k** Quantification of photon flux changes of lung metastases in (**h**) (*n* = 7 mice per group). **l, m** Representative bioluminescent images of lung metastases (**l**), brain and bone metastases (**m**) in mice treated as in (**g**) with Renca[LuM2b] and derivatives. **n** Representative X-ray images of bone metastases in mice treated as in (**g**). **o** Quantification of photon flux changes of lung metastases in (L) (*n* = 7 mice per group). **p** Quantification of photon flux of brain and bone metastases at 15 days after I.C. inoculation with indicated cells (*n* = 7 mice per group). Boxes represent data within the 25th to 75th percentiles, n.s., not significant. Immunoblots are representative of three biological replicates. In **k** and **o**, the data are presented as mean ± SEM, in (**p**), the data are presented as whisker plots: midline, median; box, 25–75th percentile; whisker, minimum to maximum values. Two-tailed Student's unpaired *t*-test were used for statistical analysis in all panels. Source data are provided as a Source data file.

(Supplementary Fig. 4g–j). We also exploited a doxycycline (Dox)-inducible (Tet-on) system to induce LAPTM5 expression in Renca[luci] cells after extravasation in the lung (Supplementary Fig. 4k, l), LAPTM5 also promoted the formation of metastases in the lung in the Tet-on system (Supplementary Fig. 4l–o), suggesting that LAPTM5 may induce metastasis initiation and tumor outgrowth in the lung stromal.

Given the proposed connection between stemness and metastasis initiation[7,8], we next investigated whether LAPTM5 promotes lung metastasis by regulating cancer stem cell (CSC)-like traits of RCC cells. First, we performed the 3D tumor sphere formation assay and found that LAPTM5-overexpressing Renca or 786-O cells formed significantly more tumor spheres than the control cells while depletion of *Laptm5* suppressed the ability of Renca[LuM2b] cells to form tumor spheres (Fig. 3e, f and Supplementary Fig. 4p), indicating the involvement of LAPTM5 in self-renewal. Next, we examined whether LAPTM5 could enhance tumor initiation in vivo. As expected, LAPTM5 overexpression significantly enhanced the subcutaneous tumor initiation ability of Renca cells in mice (Fig. 3g, h and Supplementary Fig. 5a). Conversely, silencing of *Laptm5* markedly inhibited the subcutaneous tumor initiation ability of Renca[LuM2b] cells (Fig. 3g, i and Supplementary Fig. 5a). Similarly, the tumor initiation ability of Renca cells in mouse renal subcapsular was also promoted by LAPTM5 overexpression when limited cells were implanted (Supplementary Fig. 5b). Moreover, three of the four most common embryonic stem cell (ESC) transcription factors, *NANOG*, *OCT4*, *SOX2*, and *KLF4*, which are often reactivated in aggressive and metastatic RCC[34], were significantly elevated in LAPTM5-overexpressed Renca and 786-O cells (Fig. 3j–k). Taken together, these results suggested that LAPTM5 may promote and sustain stem cell-like traits in RCC cells.

**LAPTM5 suppresses BMP signaling**. To explore the potential mechanism by which LAPTM5 regulates the CSC traits and mediates lung-specific metastasis, we first performed gene set enrichment analysis (GSEA) in The Cancer Genome Atlas (TCGA) database for two major pathological types of RCC: clear-cell RCC (KIRC, 537 samples) and papillary RCC (KIRP, 291 samples). Since the majority of LAPTM5 literature reported its negative regulation of membrane receptors[35–37], we emphasized on signaling pathways that are suppressed by LAPTM5. We found that BMP signaling was the only pathway that exhibited a significantly negative correlation with LAPTM5 levels in both KIRC and KIRP patients (Fig. 4a–c and Supplementary Data 3,

4). Intriguingly, bioactive BMP molecules from the lung stroma have been reported to inhibit the CSC traits of metastasis-initiating breast cancer cells circulating to the lung through phosphorylation of Smad 1/5/9 (Smad 1/5/8 in mice)[13,38]. Consistent with previous reports, we also detected high levels of Smad 1/5/8 phosphorylation in lung tissues, but not in the bone or brain tissue of Renca-bearing mice (Supplementary Fig. 6a). Hence, we focused on the BMP pathway. We first showed that recombinant murine BMP4 could induce the phosphorylation of Smad 1/5/8 in the parental Renca[luci] cells or the bone and brain derivatives but not the lung derivatives (Supplementary Fig. 6b and Fig. 4d). IHC analysis also confirmed the absence of Smad 1/5/8 phosphorylation in lung metastases by Renca[luci] (Fig. 4e). Based on these observations, we speculated that LAPTM5 enables RCC cells to overcome the inhibitory effect of the lung-derived BMP anti-metastatic signal on CSC traits. To prove this hypothesis, we first showed that LAPTM5-overexpressing Renca cells exhibited reduced susceptibility to BMP4-induced Smad 1/5/8 phosphorylation (Fig. 4f). Consistently, depletion of *Laptm5* rescued the resistance to BMP4 in Renca[LuM2b] cells (Fig. 4f). Similarly, in 786-O cells, phosphorylation of Smad 1/5/9 was also significantly suppressed with ectopic LAPTM5 expression upon stimulation with BMP4 (Supplementary Fig. 6c). Besides, we assessed the effect of LAPTM5 on TGF-β signaling, a pathway that interacts with BMP signals[39]. In 786-O and Renca cells, alteration of LAPTM5 expression did not affect the stimulation of Smad pathways by TGF-β1 (Supplementary Fig. 6d). Moreover, treatment of Renca cells with 30 ng/mL BMP4 for 7 days markedly inhibited tumor sphere formation which was reversed by overexpression of LAPTM5 (Fig. 4g). Conversely, treatment of Renca[LuM2b] cells with DMH1, a small molecule inhibitor of the BMP receptors, reversed the inhibitory effect of *Laptm5* knockdown on tumor sphere formation (Supplementary Fig. 6e and Fig. 4h), suggesting the function of LAPTM5 was BMP-dependent. Furthermore, the mRNA levels of *Nanog*, *Oct4* and *Sox2* in Renca or 786-O cells were inhibited by BMP4 and rescued by LAPTM5 expression (Fig. 4i and Supplementary Fig. 6f). Taken together, these data indicated that LAPTM5 enhances the stem cell-like traits of RCC cells by suppressing the activation of BMP signaling in the lung microenvironment.

**LAPTM5 downregulates BMPR1A at the post-transcriptional level**. It has been reported that BMPs, like other TGF-β family members, elicit their effects through two types of serine-threonine kinase transmembrane receptors, type I and type II BMP receptors (BMPRs)[39]. Previous studies also suggested that LAPTM5

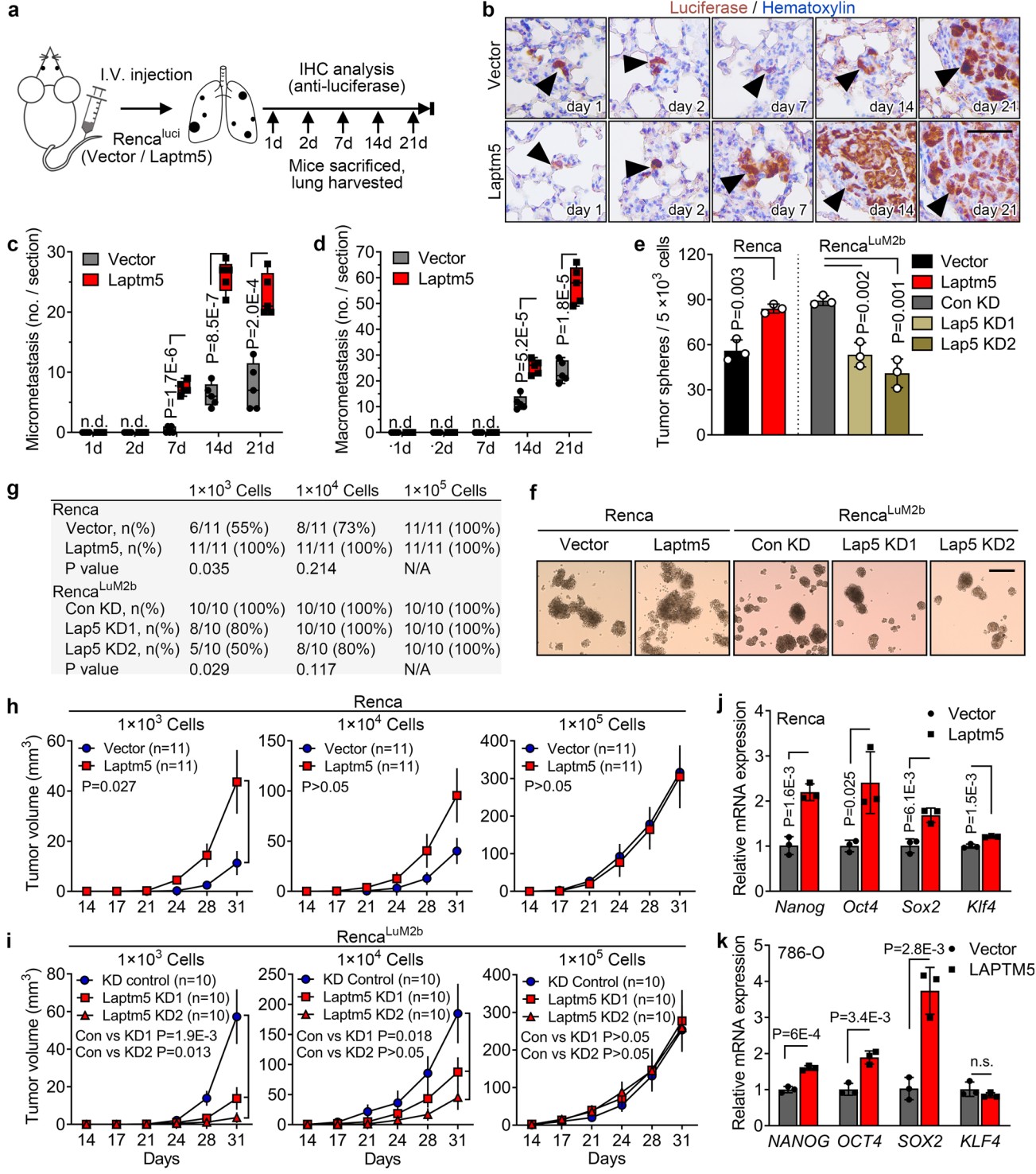

**Fig. 3 LAPTM5 promotes self-renewal and cancer stem cell traits of RCC cells. a** Schematic illustration for the steps of cell inoculation and lung metastases detection. I.V., intravenous. **b** Representative IHC images of luciferase staining in lung sections harvested from mice treated as in (**a**). Scale bar, 50 μm. Quantification of micrometastasis (diameter < 100 μm, **c**) and macrometastasis (diameter ≥ 100 μm, **d**) per lung section in (**b**) ($n = 5$ mice per group). Tumor sphere assay (**e**) and quantification of tumor sphere formation of indicated cells (**f**, $n = 3$ per group). Scale bar, 200 μm. **g** Frequencies of tumors formation in indicated groups. Tumor volumes of control and Laptm5-overexpressing Renca cells (**h**, $n = 11$ mice per group) or control and Laptm5-silenced Renca^LuM2b cells (**i**, $n = 10$ mice per group) inoculated subcutaneously with indicated cell number. **j**, **k** qRT-PCR analysis of stemness markers in control and LAPTM5-overexpressing Renca (**j**) and 786-O cells (**k**). $n = 3$ per group. In **c**, **d**, the data are presented as whisker plots: midline, median; box, 25-75th percentile; whisker, minimum to maximum values, in (**e**), (**j**), and (**k**), the data are presents as mean ± SD, in (**h**) and (**i**), the data are presents as mean ± SEM. Two-tailed Student's unpaired *t*-test was used for statistical analysis in (**c**)–(**e**), (**h**)–(**k**), $\chi^2$ test for (**g**). Source data are provided as a Source data file.

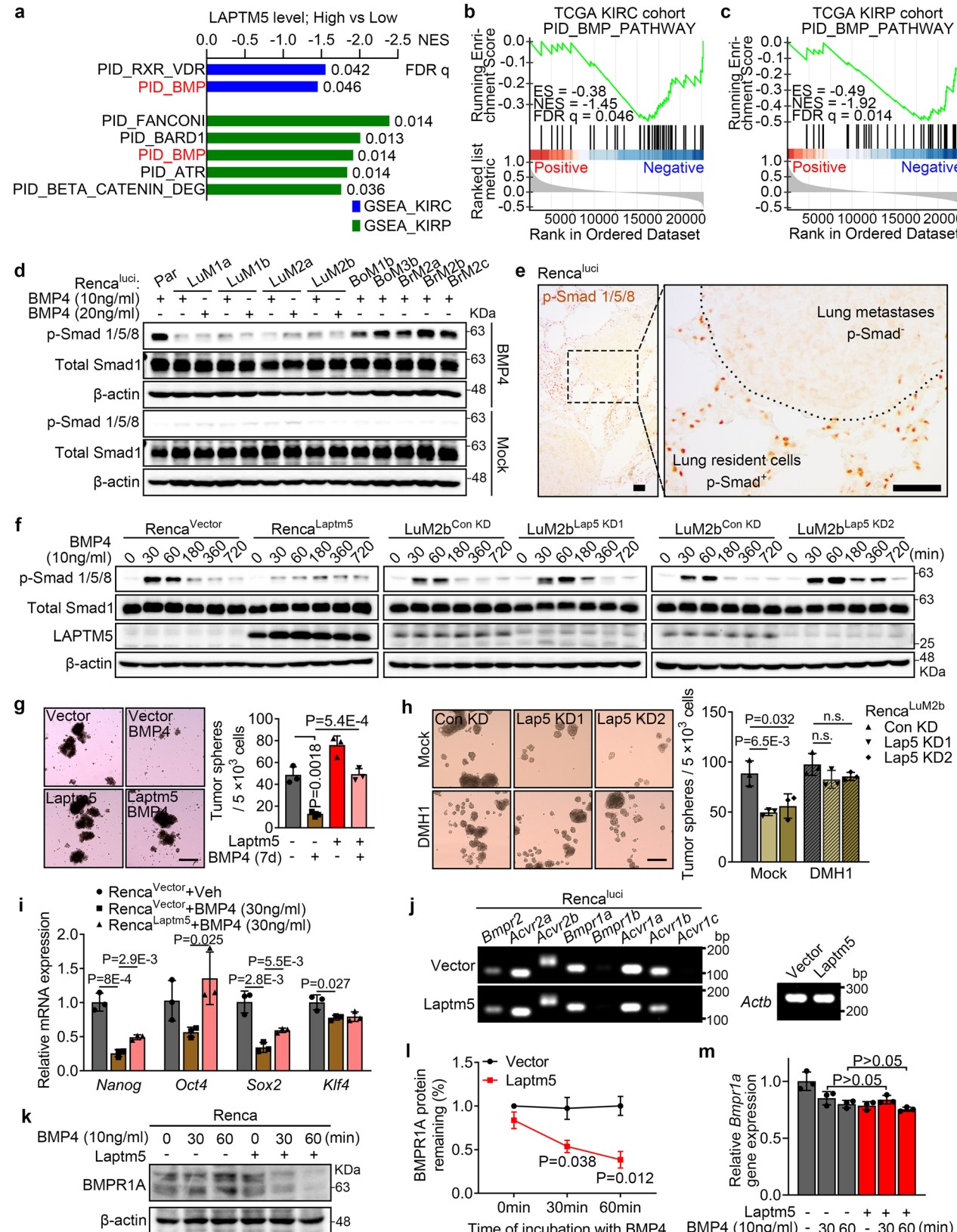

could regulate certain membrane receptor levels to modulate the response of cells to external stimuli[35,36]. Hence, we asked whether LAPTM5 could regulate BMPRs in RCC cells. We first identified the types of BMPRs expressed in RCC cells. Semiquantitative RT-PCR showed that all major known BMPRs except *Bmpr1b* and *Acvr1c* were expressed in Renca cells (Fig. 4j), and that the

transcriptional levels of these BMPRs were not affected by LAPTM5 manipulation (Fig. 4j and Supplementary Fig. 6g). However, upon treatment with 10 ng/mL murine BMP4, the protein level of one of the BMPRs, BMPR1A, gradually decreased over time in LAPTM5-overexpressing Renca cells but not in control cells, while the mRNA level of *Bmpr1a* was not affected

**Fig. 4 LAPTM5 blocks BMP signal and negatively regulates BMPR1A. a** Differentially expressed cancer-related gene sets (C2_PATHWAYS) with high LAPTM5 expression in the TCGA KIRC cohort and KIRP cohort (Top 5). NES, normalized enrichment score. FDR q, false discovery rate q value. GSEA output of genes in the PID_BMP_Pathway by LAPTM5 high and low expression groups from the KIRC cohort (**b**) and KIRP cohort (**c**) in the TCGA database. ES, enrichment score. **d** IB analysis of Renca[luci] parental and derivative cells treated with Mock (ddH2O), or recombinant murine BMP4 (10 ng/mL or 20 ng/mL) for 60 min. **e** IHC analysis of p-Smad 1/5/8 levels in lung metastases of Renca[luci]. Scale bar, 50 μm. **f** IB analysis of control and Laptm5-overexpressing Renca cells (left panel) or control and Laptm5-silenced Renca[LuM2b] cells (middle and right panel) treated with BMP4 (10 ng/mL) for indicated times. **g** Representative images of tumor sphere assay of control and Laptm5-overexpressing Renca cells treated with or without BMP4 (30 ng/mL) for 7 days, and quantification of tumor sphere formation of indicated groups (n = 3 per group). Scale bar, 200 μm. **h** Representative images of tumor sphere assay of control and Laptm5-silenced Renca[LuM2b] cells treated with or without DMH1 (5 μM) for 7 days, and quantification of tumor sphere formation of indicated groups (n = 3 per group). Scale bar, 200 μm. **i** qRT-PCR analysis of stemness markers in indicated cells. n = 3 per group. **j** Semiquantitative RT-PCR analysis of BMP receptors in control and Laptm5-overexpressing Renca[luci] cells with *Actb* as control. **k** IB analysis of BMPR1A in control and Laptm5-overexpressing Renca cells treated with BMP4 (10 ng/mL) for indicated times. **l** Ratio of BMPR1A protein remaining in control and Laptm5-overexpressing Renca cells treated as in (**k**) (n = 3 per group). **m** qRT-PCR analysis of *Bmpr1a* in control and Laptm5-overexpressing Renca cells treated with BMP4 (10 ng/mL) for indicated times. n = 3 per group. Immunoblots are representative of three biological replicates. Data in bar graphs are presented as mean ± SD. Two-tailed Student's unpaired t-test were used for statistical analysis in all panels. Source data are provided as a Source data file.

(Fig. 4k–m and Supplementary Fig. 6h, i). Correspondingly, BMPR1A protein level was upregulated in *Laptm5*-silenced Renca[LuM2b] cells (Supplementary Fig. 6j). Similarly, LAPTM5-overexpressing 786-O cells also showed a decrease in BMPR1A protein level but not mRNA level 60 min after BMP4 treatment (Supplementary Fig. 6k, l). Together, these data suggested that LAPTM5 negatively regulated BMPR1A via post-transcriptional mechanisms in RCC cells.

**LAPTM5 promotes BMPR1A lysosomal sorting and binding with WWP2.** To investigate how LAPTM5 mediates the down-regulation of BMPR1A protein, we stably overexpressed Flag-tagged LAPTM5 in 786-O and Renca cells, respectively. Proteins that could interact with LAPTM5 were then identified using IP-MS. After treatment with 10 ng/mL BMP4 for 60 min, LAPTM5-binding proteins were immunoprecipitated with Flag antibody and subjected to LC-MS/MS analysis (Fig. 5a, b and Supplementary Fig. 7a). A total of 151 and 113 proteins were identified in 786-O and Renca cells, respectively; of which 9 proteins were common in both lines, WW domain-containing E3 ubiquitin protein ligase 2 (WWP2, also called AIP2), a member of the NEDD4 protein family, ranked among the top of the list as a protein that might interact with LAPTM5 (Fig. 5b–d and Supplementary Fig. 7a). Co-immunoprecipitation experiments confirmed the interaction between LAPTM5 and WWP2 in both 786-O and Renca cells, which was further enhanced upon BMP4 treatment (Supplementary Fig. 7b, c). To determine if other NEDD4 family members also bind to LAPTM5, we carried out immunoprecipitation assays in LAPTM5-overexpressing 786-O or Renca cells and found that WWP2 was the only NEDD4 family protein that can bind to LAPTM5 (Supplementary Fig. 7d). Moreover, the interaction between LAPTM5 and WWP2 was also confirmed by immunofluorescence experiments (Supplementary Fig. 7e). NEDD4 family proteins, including SMURF1 and NEDD4L, have been shown to promote the endocytosis and/or ubiquitin-mediated degradation of membrane proteins[40,41]. To investigate whether LAPTM5 promotes BMPR1A degradation via WWP2, we ectopically expressed Myc-tagged BMPR1A and Flag-tagged LAPTM5 or HA-tagged WWP2 in 786-O cells, immunoprecipitation assays showed that BMPR1A receptor bound to both LAPTM5 and WWP2, the binding was enhanced by BMP4 treatment, and the total BMPR1A protein level was reduced in the presence of LAPTM5 or WWP2 (Supplementary Fig. 7f, g). Evidence of co-localization between WWP2 and BMPR1A was also provided by immunofluorescence assays in 786-O cells (Supplementary Fig. 7h). In addition, we performed serial IP experiments (Myc-BMPR1A pulldown followed by HA-WWP2 pulldown) and successfully detected the Flag-LAPTM5 signal,

further proving that LAPTM5-WWP2-BMPR1A were in the same protein complex (Fig. 5e, f). Furthermore, when all three constructs were co-transfected in 293T cells or 786-O cells, LAPTM5 dramatically promoted the interaction between WWP2 and BMPR1A (Fig. 5g, h and Supplementary Fig. 7i), suggesting a crucial regulatory role of LAPTM5 in BMPR1A-WWP2 interaction. Likewise, endogenous interactions between LAPTM5-WWP2 and WWP2-BMPR1A were also confirmed in parental 786O[luci/eGFP] and the lung-met derivative 786O[LuM1a] cells (Fig. 5i).

WWP2 belongs to the NEDD4 family of E3 ubiquitin ligases that contains a conserved amino-terminal C2 domain, a variable number of WW domains, and a carboxy-terminal HECT domain[42]. To find out the specific binding sites between WWP2 and LAPTM5, we performed immunoprecipitation assays using either the full-length WWP2 construct or mutant WWP2 constructs lacking various functional domains (Fig. 5j), and found that all constructs shorter than 407–868 failed to bind with LAPTM5, suggesting that the C2 domain and the first two and the last WW domains are dispensable for LAPTM5 binding, whereas the WW3 domain is required (Fig. 5k).

It was reported that ectopically expressed LAPTM5 mainly localizes to late endosomes and/or lysosomes[43,44]. In RCC cells, we confirmed that endogenous LAPTM5 mostly co-localizes with the late endosome and/or lysosome makers LAMP1 and RAB7, but not with early endosome markers EEA1 and RAB5[45] (Supplementary Fig. 7j). Using immunofluorescence, we observed that LAPTM5 facilitated BMPR1A endocytosis in untreated and BMP4-treated 786-O cells. Moreover, BMP4 treatment induced a more dramatic increase in BMPR1A endocytosis in LAPTM5-overexpressing cells compared to control cells (Fig. 5l–n). In addition, BMPR1A and LAPTM5 displayed the most co-localization (Fig. 5m, o, green and red lines, respectively), and BMP4 treatment enhanced the co-localization signal (Fig. 5m, o, yellow line), further confirming the previous immunoprecipitation results. Similarly, in lung metastatic 786O[LuM1a] cells, we also observed an increase in the late endosome and/or lysosome formed by LAPTM5, as well as an increase in BMPR1A endocytosis, which was further enhanced by BMP4 treatment (Supplementary Fig. 7k). Taken together, these data demonstrated that LAPTM5 facilitates the endocytosis of BMPR1A and its interaction with WWP2.

**WWP2 mediates lysosome-based ubiquitination and degradation of BMPR1A.** It is well established that ubiquitination of membrane receptors is required for lysosome-mediated degradation, we, therefore, investigated whether WWP2 ubiquitylates BMPR1A. In 293T cells, the conjugation of ubiquitin chains to

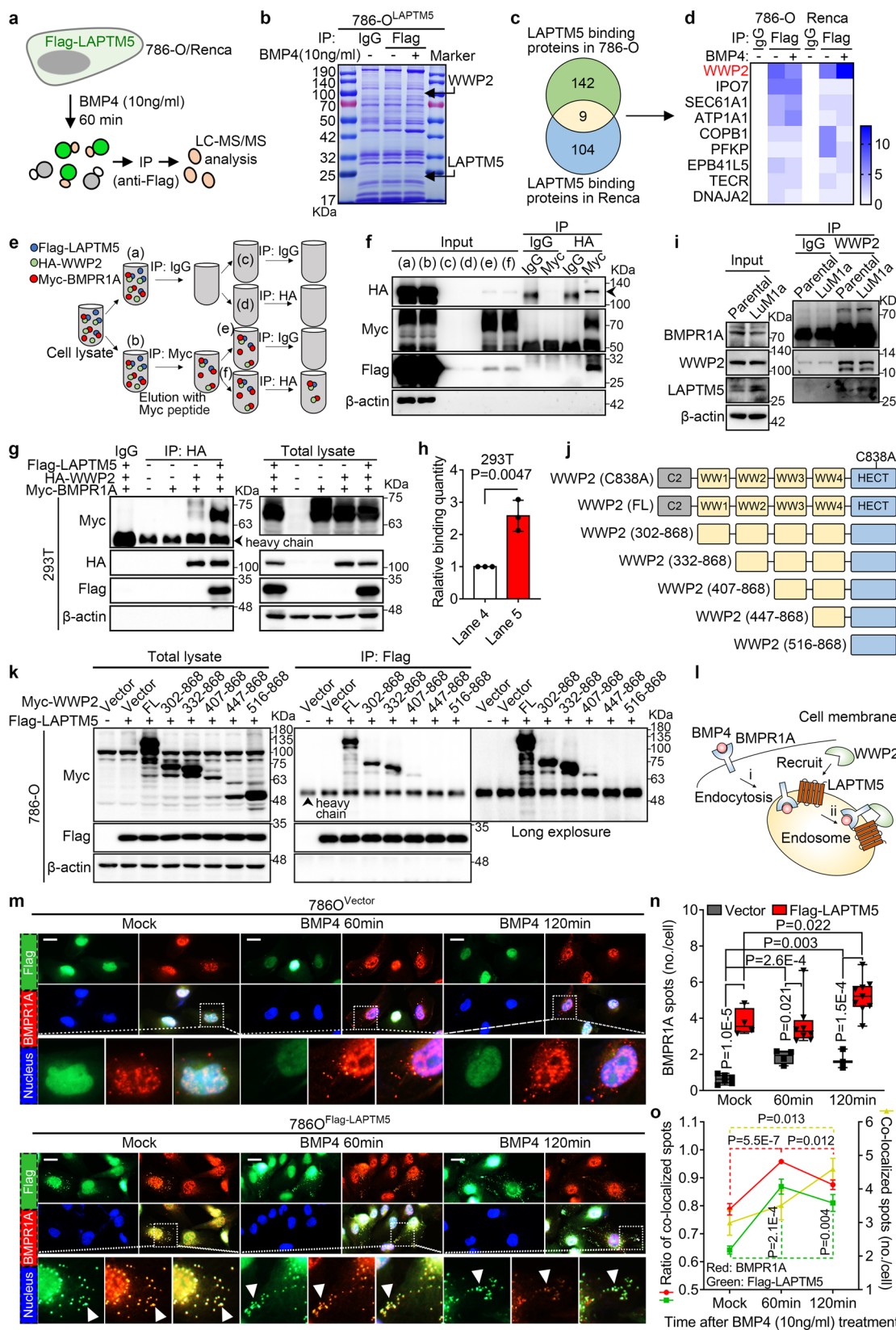

BMPR1A was markedly increased in the presence of WWP2, and further increased by the co-expression of LAPTM5 (Supplementary Fig. 8a), suggesting that LAPTM5 promoted the interaction between WWP2 and BMPR1A. The ubiquitination of BMPR1A by WWP2 was also confirmed in the RCC cell line 786-O (Fig. 6a). In both 293T and 786-O cells, BMPR1A

ubiquitination was inhibited by the expression of a catalytically inactive mutant of WWP2 (WWP2$^{C838A}$)[46] (Fig. 6b and Supplementary Fig. 8b). Moreover, silencing of WWP2 with three different small interfering RNAs (siRNA) also reduced BMPR1A ubiquitination (Fig. 6c and Supplementary Fig. 8c). Furthermore, treatment of 293T cells co-expressing LAPTM5 and

**Fig. 5 LAPTM5 directly interacts with WWP2 and promotes BMPR1A lysosomal sorting. a** Schematic illustration of TMT-labeled LC-MS/MS analysis in 786O[LAPTM5] and Renca[Laptm5] cells treated or untreated with BMP4 (10 ng/mL) for 60 min. **b** Coomassie brilliant blue staining of 786O[LAPTM5] cells treated or untreated with BMP4 (10 ng/mL) for 60 min. The arrowheads indicate bands of WWP2 and LAPTM5, respectively. **c** Overlap of identified LAPTM5-binding proteins in 786O[LAPTM5] and Renca[Laptm5] cells. **d** Heatmap of the nine proteins identified binding to LAPTM5 by TMT-labeled LC-MS/MS analysis. Schematic illustration (**e**) and IB result (**f**) of serial immunoprecipitation (IP) analysis using anti-Myc (first round) and anti-HA (second round) in 293T cells transfected with expression vectors for Flag-LAPTM5, HA-WWP2, and Myc-BMPR1A. Arrowhead indicates the band of HA-WWP2. **g** IP and IB analyses of 293T cells transfected with expression vectors for Flag-LAPTM5, HA-WWP2, and Myc-BMPR1A. **h** Quantitation of the binding between HA-WWP2 and Myc-BMPR1A as treated in (**g**) (n = 3 per group). **i** IP and IB analyses of 786O[luci/eGFP] Parental and LuM1a cells. **j** Schematic illustration of full-length (FL) WWP2 and its deletion mutants lacking functional domains (ranges indicate amino acids present in construct). C838A represents the mutation of cytosine to adenine in the 838th nucleotide. **k** IP and IB analyses of 786-O cells transfected with expression vector for Flag-LAPTM5, control (Vector) or Myc-tagged full-length WWP2 or various deletion mutants. **l** Schematic illustration of LAPTM5, WWP2 and BMPR1A interactions in cells.
**m** Immunofluorescence (IF) analysis of control (upper panel) and LAPTM5-overexpressing (lower panel) 786-O cells treated with Mock (ddH₂O), or BMP4 (10 ng/mL) for 60 min or 120 min. Scale bar, 10 μm. **n** Quantification of BMPR1A punctate spots per 786-O cell treated as in (M) [n = biological replicates, 6 in Vector, 4 in LAPTM5 (Mock); n = 4 in Vector, 8 in LAPTM5 (60 min); n = 3 in Vector, 9 in LAPTM5 (120 min)]. Data are presented as whisker plots: midline, median; box, 25–75th percentile; whisker, minimum to maximum values. **o** Ratio of co-localized spots to total BMPR1A (red line) or LAPTM5 (green line) spots and quantification of co-localized spots per 786-O cell (yellow line) treated as in (**m**) (n = 4, 8, 9 biological replicates in group Mock, 60 min, 120 min, respectively). Immunoblots are representative of three biological replicates. Data are presented as mean ± SD in (**h**), and mean ± SEM in (**o**). Two-tailed Student's unpaired t-test was used for statistical analysis in all panels. Source data are provided as a Source data file.

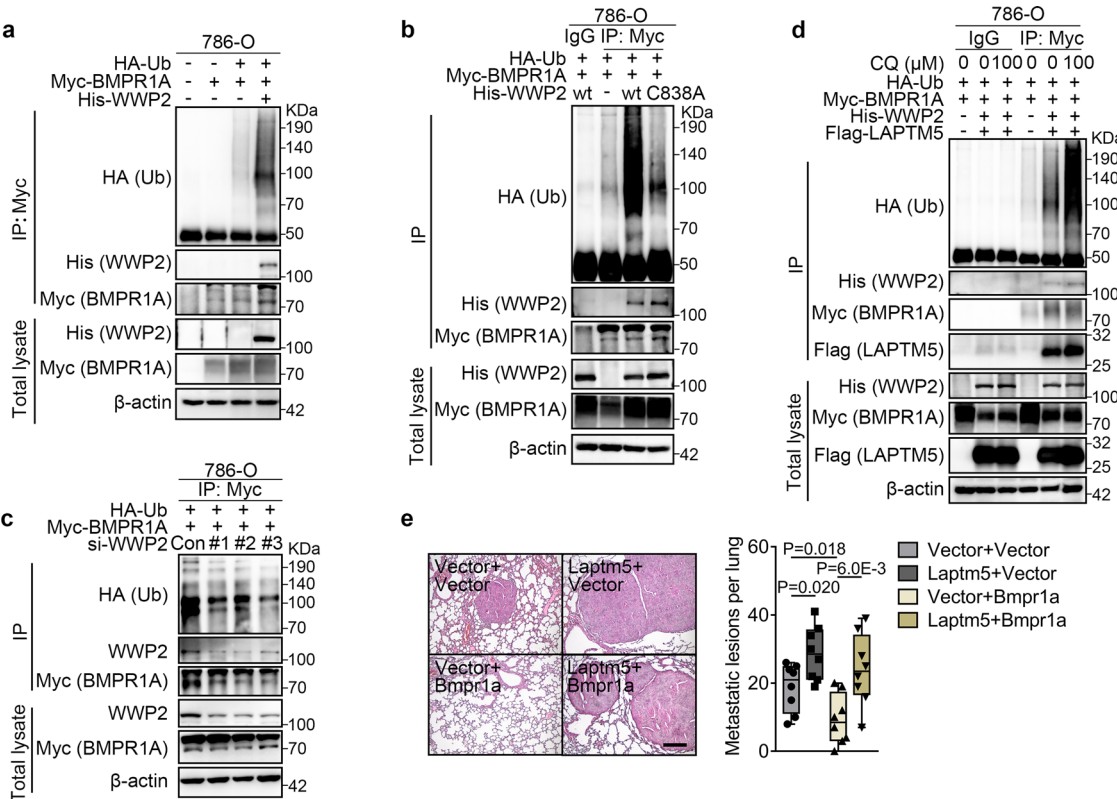

**Fig. 6 WWP2 promotes lysosome-based polyubiquitylation and degradation of BMPR1A. a** IP and IB analyses of 786-O cells transfected with expression vectors for His-WWP2, Myc-BMPR1A and HA-ubiquitin (Ub). **b** IP and IB analyses of 786-O cells transfected with expression vectors for His-tagged wild type (wt) or C838A mutated WWP2, Myc-BMPR1A and HA-Ub. **c** IP and IB analyses of 786-O cells transfected with siRNAs against WWP2 or expression vectors for Myc-BMPR1A and HA-Ub. **d** IP and IB analyses of 786-O cells transfected with expression vectors for Flag-LAPTM5, His-WWP2, Myc-BMPR1A, HA-Ub and treated with different concentrations of chloroquine (CQ) for 60 min. **e** Representative H&E images of lung metastases at 21 days after I.V. injection of Renca cells stably expressing Laptm5 and/or Bmpr1a (left panel) and quantification of the lung metastatic lesions per slide (right panel, n = 8 mice per group). Data are presented as whisker plots: midline, median; box, 25–75th percentile; whisker, minimum to maximum values. Two-tailed Student's unpaired t-test was used for statistical analysis. Immunoblots are representative of three biological replicates. Source data are provided as a Source data file.

WWP2 with the lysosome inhibitor chloroquine diphosphate (CQ) resulted in a dose-dependent increase in both BMPR1A ubiquitination and total BMPR1A protein levels (Fig. 6d and Supplementary Fig. 8d), supporting that LAPTM5-based lysosome pathway promoted the degradation of ubiquitinylated BMPR1A. Lastly, in the in vivo lung metastasis assay using Renca cells, the metastatic suppressor role of ectopically expressed BMPR1A was reversed by LAPTM5 co-expression (Fig. 6e). Taken together, these data revealed that WWP2 facilitated BMPR1A polyubiquitination and that the LAPTM5/WWP2-based lysosome pathway mediated the integrated process of BMPR1A degradation.

**LAPTM5 negatively correlates with BMPR1A in clinical RCC specimens and predicts lung metastasis.** To study the expression patterns of LAPTM5 and BMPR1A in clinical samples, we gathered a cohort of 34 RCC metastases from the lung, bone, and brain, as well as 106 normal kidney tissues and 150 primary RCC tissues and performed IHC analysis for both LAPTM5 and BMPR1A. LAPTM5 displayed remarkably higher expression levels in lung metastases than in bone ($P < 0.001$) and brain metastases ($P = 0.039$), and primary RCC tissues ($P < 0.0001$) (Fig. 7a, b). We also confirmed that neither bone nor brain metastases showed divergent expression of LAPTM5 compared with primary tumors (Fig. 7a, b). More importantly, BMPR1A expression levels were lower in lung metastases than in other organ metastases (Fig. 7c and Supplementary Fig. 9a). Quantitation of LAPTM5 and BMPR1A expression levels showed a tight negative correlation pattern in RCC metastases ($R = -0.459$, $P = 0.006$) (Fig. 7d).

To evaluate the relationship between LAPTM5 and BMPR1A expression in primary RCC tumors, we collected and analyzed clinical information from patients with primary RCC tumors. Histological staining showed that LAPTM5 was upregulated in RCC tissues compared with normal kidney tissues ($P < 0.0001$) (Fig. 7a, b). Besides, staining of two consecutive tissue sections with either LAPTM5 or BMPR1A antibody showed high regional LAPTM5 expression in most primary RCC tissues, as well as a negative correlation between LAPTM5 and BMPR1A expression (Supplementary Fig. 9b, c), which is similar to that observed in RCC metastases. Moreover, statistical analysis revealed a significant correlation between the staining intensity of LAPTM5 and histological grade ($P = 0.039$), TNM stage ($P = 0.038$), adverse pathological events ($P = 0.002$) and distant metastasis ($P < 0.001$) of primary RCC (Supplementary Fig. 9d and Supplementary Data 5). In addition, primary RCC tissues of patients with lung metastases displayed higher LAPTM5 levels than those without lung metastases but lower LAPTM5 levels than lung metastases tissues (Fig. 7e and Supplementary Fig. 9e). To further assess the predictive value of LAPTM5 expression for RCC lung metastasis, we performed univariate and multivariable logistic regression analyses. The results revealed that LAPTM5 expression, like the histological grade, was an independent predictor of lung metastasis in RCC ($P < 0.001$) (Fig. 7f). Furthermore, reduced metastatic-free survival (MFS) and overall survival (OS) were found in RCC patients with high LAPTM5 expression in primary tumors (Fig. 7g, h). These clinical data confirmed the negative correlation between LAPTM5 and BMPR1A as depicted in the schematic (Fig. 7i). LAPTM5 expression levels may thus serve as an independent prognosis factor of lung metastasis and survival of RCC patients.

**LAPTM5 is specifically activated in lung metastasis of multiple cancers.** To extend our findings to other cancer types, we analyzed the transcriptional profile of a set of breast cancer cell lines generated by in vivo organotropic metastatic clone selection[19] (Fig. 8a). In this dataset, LAPTM5 also exhibited higher levels in breast cancer cells with high lung-specific metastatic activity (Lung-M) than in those with mediate activity (Mediate) and bone-specific metastatic activity (Bone-M) (Fig. 8b). Similarly, when we examined LAPTM5 expression in a melanoma dataset (GSE50496) and a colorectal cancer dataset (GSE41258), higher levels of LAPTM5 were also found in lung metastases than other organ metastases[47,48] (Fig. 8c, d).

In agreement with the findings in RCC cells, overexpression of LAPTM5 promoted the sphere formation ability of the murine 4T1[luci] and human MDA-MB-231 breast cancer cell lines in vitro while silencing of *Laptm5* suppressed sphere formation (Fig. 8e and Supplementary Fig. 10a–d). LAPTM5 also increased the ability of

4T1[luci] cells to form orthotopic tumors in vivo when implanted into the mammary glands of mice (Fig. 8f, g and Supplementary Fig. 10e, f). More importantly, under the condition ($1 \times 10^5$ cells) where no difference in primary tumor formation rate was observed (Fig. 8g and Supplementary Fig. 10e, f), LAPTM5 overexpression resulted in increased metastatic lesion-forming activity in the lung, and *Laptm5* knockdown significantly inhibited lung metastasis (Fig. 8f, h, i).

Next, we examined whether LAPTM5 affected the response of breast cancer cells to BMP stimulation. Immunoblotting results showed that LAPTM5 silencing accelerated the phosphorylation of Smad 1/5/9 in MDA-MB-231 cells in response to BMP4 treatment (Supplementary Fig. 10g). Immunoprecipitation assay also confirmed the interaction between LAPTM5 and WWP2 in 4T1[luci] cells; which was enhanced in the presence of BMP4 (Supplementary Fig. 10h). LAPTM5 and BMPR1A also exhibited co-localization in MDA-MB-231[LAPTM5] cells (Supplementary Fig. 10i). After pretreatment with 10 ng/mL BMP4 for 60 or 120 min, the punctate spots formed by BMPR1A in cells increased significantly and LAPTM5 overexpression promoted the endocytosis of BMPR1A in MDA-MB-231 cells (Supplementary Fig. 10j, k). Moreover, WWP2 was also found to bind to BMPR1A and promote its degradation in MDA-MB-231 cells (Supplementary Fig. 10l).

In conclusion, these data demonstrated that the activation of LAPTM5 in lung metastases is a common molecular event shared by several types of human cancer, and that the LAPTM5/WWP2-based lysosomal regulatory pathway mediates the ubiquitination and degradation of BMPR1A.

## Discussion

Organ specificity observed in cancer metastasis is known as organotropism and remains one of the most intriguing and unanswered questions in cancer biology. With regard to lung-specific metastasis, limited but essential efforts have been devoted to breast cancer and only a few groundbreaking discoveries have been made[13,19,38]. However, related research on RCC has not yet been reported. Here, we document a critical role of LAPTM5, which regulates the self-renewal and cancer stem cell-like traits of RCC cells in lung stroma, thus promoting lung-specific metastasis of RCC.

By integrated analyses of transcriptional profiles in clinical samples and Renca[luci] RCC cell line derivatives with distinct organ metastasis tendency, we showed that LAPTM5 was preferentially activated in lung metastasis and was required for RCC cells to develop lung-specific metastases. Phenotypically, we founded that LAPTM5 contributed to the self-renewal and cancer stem cell-like traits of RCC cells in lung stroma by blocking the effect of lung stroma-derived BMPs. Mechanistic studies using IP-MS in murine and human RCC cell lines (Renca and 786-O) uncovered aLAPTM5 binding protein WWP2 and established the regulatory role of LAPTM5/WWP2 in the ubiquitination and lysosomal degradation of BMPR1A. Furthermore, in primary and metastatic RCC clinical samples, we showed that LAPTM5 negatively correlated with BMPR1A levels and predicted lung metastatic frequency in renal cancer. Most importantly, activation of LAPTM5 in lung metastasis is a common molecular event shared by several types of cancer types. Another possibility that was not investigated in the current study is the tight correlation between LAPTM5 and immune-related or chemotactic pathways, like T/B cell antigen receptor (TCR/BCR)- and interleukin (IL)-associated pathways, in RCC. LAPTM5 has been reported to negatively regulate cell surface TCR/BCR expression via promoting their degradation in the lysosomal compartments to mediate T and B lymphocyte inactivation[35,36]. Moreover,

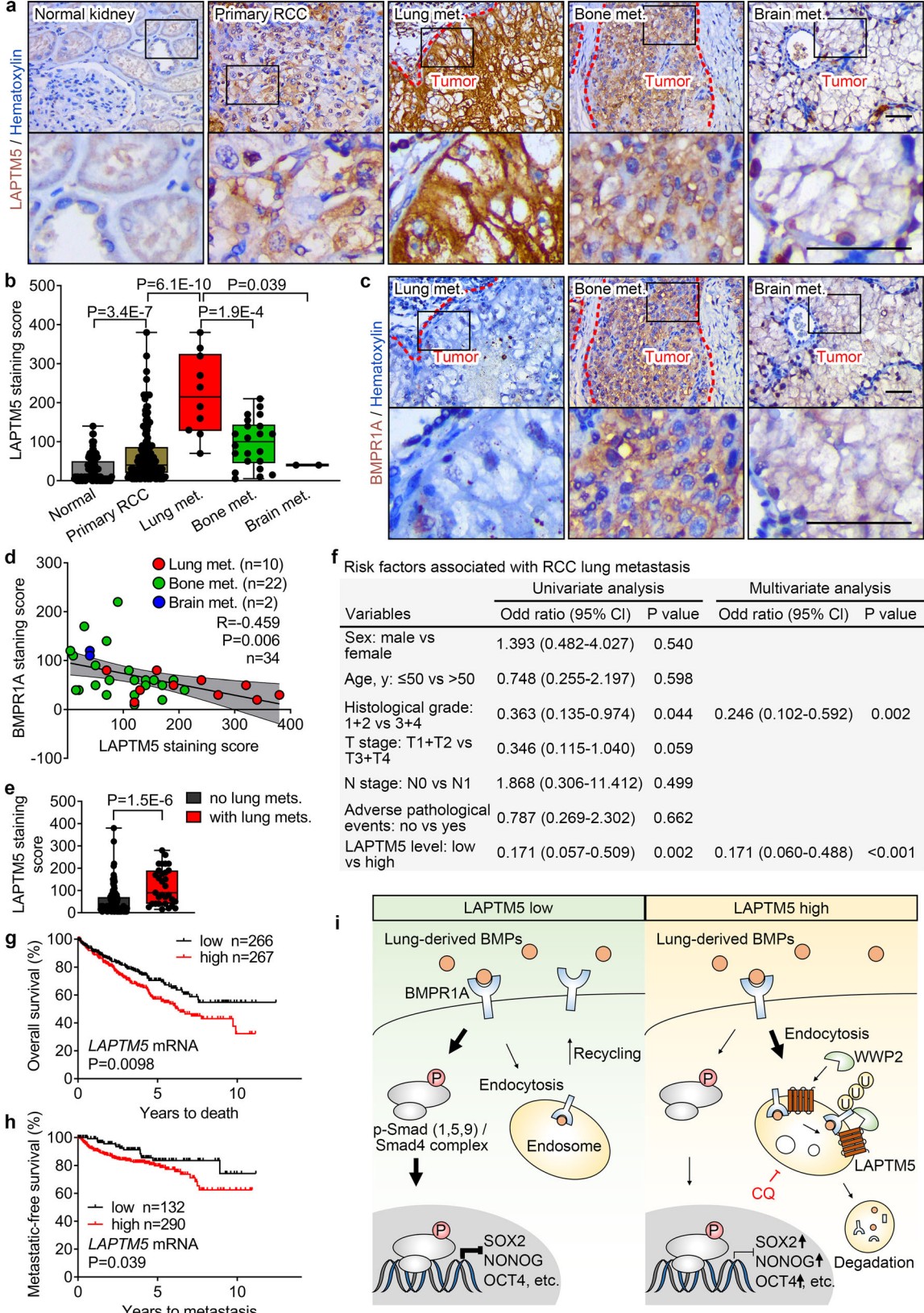

LAPTM5 may be involved in regulating the immune microenvironment in KIRC[49]. Whether LAPTM5 could also mediate lung-specific metastasis by regulating infiltrated immune cell function in primary or metastatic foci warrants further investigation.

Controversial roles of BMPs in organotrophic metastasis have been reported. For example, one study found that blocking BMP signal could lead to lung-specific metastasis in breast cancer[13], while another study indicated that BMP signaling enhances bone metastasis of breast cancer through the Smad pathway[50].

**Fig. 7 LAPTM5 negatively correlates with BMPR1A and predicts lung metastasis of RCC. a** Representative IHC images for LAPTM5 in clinical sections of normal kidney tissue, primary RCC tissue, lung metastases (Lung met.), bone metastases (Bone met.) and brain metastases (Brain met.) from RCC patients. Scale bar, 100 μm. **b** IHC staining score of LAPTM5 in clinical primary RCC and organ metastases from RCC patients. Normal, $n = 106$ samples; Primary RCC, $n = 150$ samples; Lung met., $n = 10$ samples; Bone met., $n = 22$ samples; Brain met., $n = 2$ samples. **c** IHC images for BMPR1A in clinical sections [Serial section of (**a**)] of lung metastases (Lung met.), bone metastases (Bone met.) and brain metastases (Brain met.) from RCC patients. Scale bar, 100 μm. **d** Pearson correlation between LAPTM5 and BMPR1A levels in clinical RCC metastases. R, Pearson correlation coefficient; center line, mean of best-fit line; the shadow indicates 95% confidence interval. **e** IHC staining score of LAPTM5 in primary RCC patients with lung metastases (with lung met., $n = 31$ samples) and those without lung metastases (no lung met., $n = 119$ samples). **f** Univariate and multivariate analyses to determine risk factors associated with lung metastasis of RCC patients ($n = 150$ patients). CI: confidence interval. Kaplan–Meier survival curves for overall survival (**g**) and metastatic-free survival (**h**) of patients with low and high *LAPTM5* mRNA level in the TCGA KIRC cohort. **i** Schematic illustration of LAPTM5 promoting lung-specific metastasis of RCC. CQ, chloroquine; P, phosphorylated; U, ubiquitylated. In **b** and **e**, the data are presented as whisker plots: midline, median; box, 25–75th percentile; whisker, minimum to maximum values. Two-tailed Student's unpaired t-test in (**b**) and (**e**). Source data are provided as a Source data file.

We reasoned that BMPs play different roles in different tissues and that each organ uses distinct signaling molecules to suppress cancer metastasis. Thus, although BMPs are also present in the bone microenvironment because of its essential role in bone development and turnover[39,51,52], it appears not to be used by the bone to suppress metastatic colonization of circulating tumor cells. In contrast, the lung tissues, at least in our models, exhibited much higher expression of BMPs, which appears to be the dominant signal used by the lung stroma to suppress the outgrowth of metastatic tumor cells. Moreover, our data also suggest that this metastasis suppressor role of BMP in the lung is not confined to renal cancer but a common molecular event shared by several types of human cancer.

The metastasis process involves the evolution, dissemination, and subsequent colonization and exit from dormancy of cancer cells from a primary tumor to a distant organ[32,53]. Elevated LAPTM5 expression in lung metastases may occur at every phase of metastasis. According to our finding that higher LAPTM5 levels were found in primary RCC tissues with lung metastases than in those without lung metastases, we inferred that LAPTM5 is upregulated in primary tumors and is responsible for the initiation of lung metastasis. However, unlike in the Jon_Re-nal_Cancer dataset[30], we detected even higher LAPTM5 levels in the lung metastases tissues than primary tumor tissues in patients with lung metastases (Supplementary Fig. 9e); besides, LAPTM5 level was increased after TGF-β treatment[54], therefore, whether the lung indigenous TGF-β-rich environmental causes the upregulation of LAPTM5 remains to be confirmed. In favor of the former hypothesis, most tumor cells in primary lesions did not show homogeneous LAPTM5 expression, but exhibited regionally high LAPTM5 expression[26]. However, the mechanisms underlying LAPTM5 activation in these cells remain unexplored.

After the surgical resection of primary tumors, 17.5–21% RCC patients develop local recurrence or distant metastasis with the lung being the preferred organ[55,56]. Histological analysis of LAPTM5 in postoperative tissues, as revealed in this work, might serve as a predictor of lung metastasis in RCC patients and be helpful for follow-up plans and treatment decisions. More importantly, the insights from this work suggest a different translational strategy. Lysosome inhibitors, including CQ, hydroxychloroquine (HCQ), and its novel derivatives ROC325 and Lys05, either alone or in combination therapy, have shown excellent anti-tumor activity in preclinical models[57–62]. Although in previous clinical trials, the therapeutic effects of most lysosome inhibitors were disappointing against various cancers such as pancreatic cancer, colon cancer, glioma, and breast cancer[63–67], the potent effect of CQ in blocking LAPTM5 and restoring BMPR1A levels makes it a promising therapeutic agent for patients with lung metastasis. Development of specific small-molecule inhibitors for LAPTM5 is also needed. Moreover,

lysosomal autophagy participates in sunitinib resistance in RCC[68] and immune evasion in pancreatic cancer[69]; preliminary attempts to combine targeted or immune checkpoint inhibitors with lysosome inhibitors in the treatment of RCC might lead to new directions in the control of advanced tumors[70,71], especially those with lung metastasis.

In summary, our work herein revealed the critical role of LAPTM5, a lysosome transmembrane protein, in lung-specific metastasis. LAPTM5 recruits WWP2 and mediates the ubiquitination, lysosomal sorting, and degradation of the critical BMP receptor BMPR1A, thereby interfering with lung stroma-derived anti-metastatic BMP signals and ultimately promotes lung-specific metastasis. Our results suggested that LAPTM5 may be a potential therapeutic target for lung metastasis of multiple cancer types.

## Methods
**Cell culture.** 786-O, 293T, 4T1, and MDA-MB-231 cells were obtained from the Cell Bank of the Chinese Academy of Science (Shanghai, China). Renca cells were purchased from ATCC. Cell line identity was confirmed by fingerprinting (Bio-Research Innovation Center, Suzhou, China). Renca and 4T1 cells were transduced with a lentivirus expressing firefly luciferase, 786-O cells were transduced with a lentivirus expressing enhanced GFP (eGFP) and firefly luciferase. Puromycin was added to sort luciferase and eGFP positive cells. 786-O, Renca, and 4T1 cells were cultured in RPMI 1640 medium with 10% fetal bovine serum (FBS). 293T and MDA-MB-231 cells were maintained in DMEM medium with 10% FBS. All complete mediums were supplemented with 100 U/mL penicillin and 100 μg/mL streptomycin. Cultures were maintained at 37 °C under a humidified atmosphere containing 5% $CO_2$.

**Plasmid, lentivirus construction, and transfection.** Small interfering RNAs (siRNAs) against human *LAPTM5* and *WWP2* were purchased from GenePharma (Shanghai, China). Validated sequences were cloned into the vector pLKO.1-Puro to construct short hairpin RNAs (shRNAs). Flag-tagged human LAPTM5 and His-tagged catalytically inactive mutant of WWP2 (cysteine in the 838th amino acid was replaced with alanine, WWP2$^{C838A}$) vectors were purchased from OBiO Technology (Shanghai, China). HA-tagged ubiquitin was obtained from Addgene. Murine *Laptm5* and *Bmpr1a* coding sequences were cloned into the expression vector pCDH-CMV-Flag-EF1-Puro, while Myc-tagged human BMPR1A, HA-tagged murine WWP2, HA-/His-tagged full-length and mutant truncated human WWP2 were cloned and expressed in a pcDNA3.1(-)-Puro vectors. Plasmids were transfected into cells using Lipofectamine 2000 per manufacturer's instructions.

TET-inducible Flag-Laptm5/Control lentivirus were obtained from GeneChem (Shanghai, China). To produce lentivirus, 5 μg pLP1, 2.5 μg pLP2, 5 μg pLP/VSVG plasmids, and 5 μg vectors expressing control shRNA, specific shRNA constructs (against murine LAPTM5), or human and murine LAPTM5 were transfected into 293T cells with 70–80% confluence in 100 mm plate using Lipofectamine 2000. After incubation for 48–72 h, the supernatants containing lentivirus were harvested and used to infect target cells with polybrene for 12 h in the incubator. After selection with puromycin for 7 days, cells were harvested to determine the knockdown or overexpression efficiency.

**Mouse studies and isolation of lung-derived cells.** All animal studies were conducted according to the guidelines of the Institutional Animal Care and Use Committee of Nanjing Drum Tower Hospital.

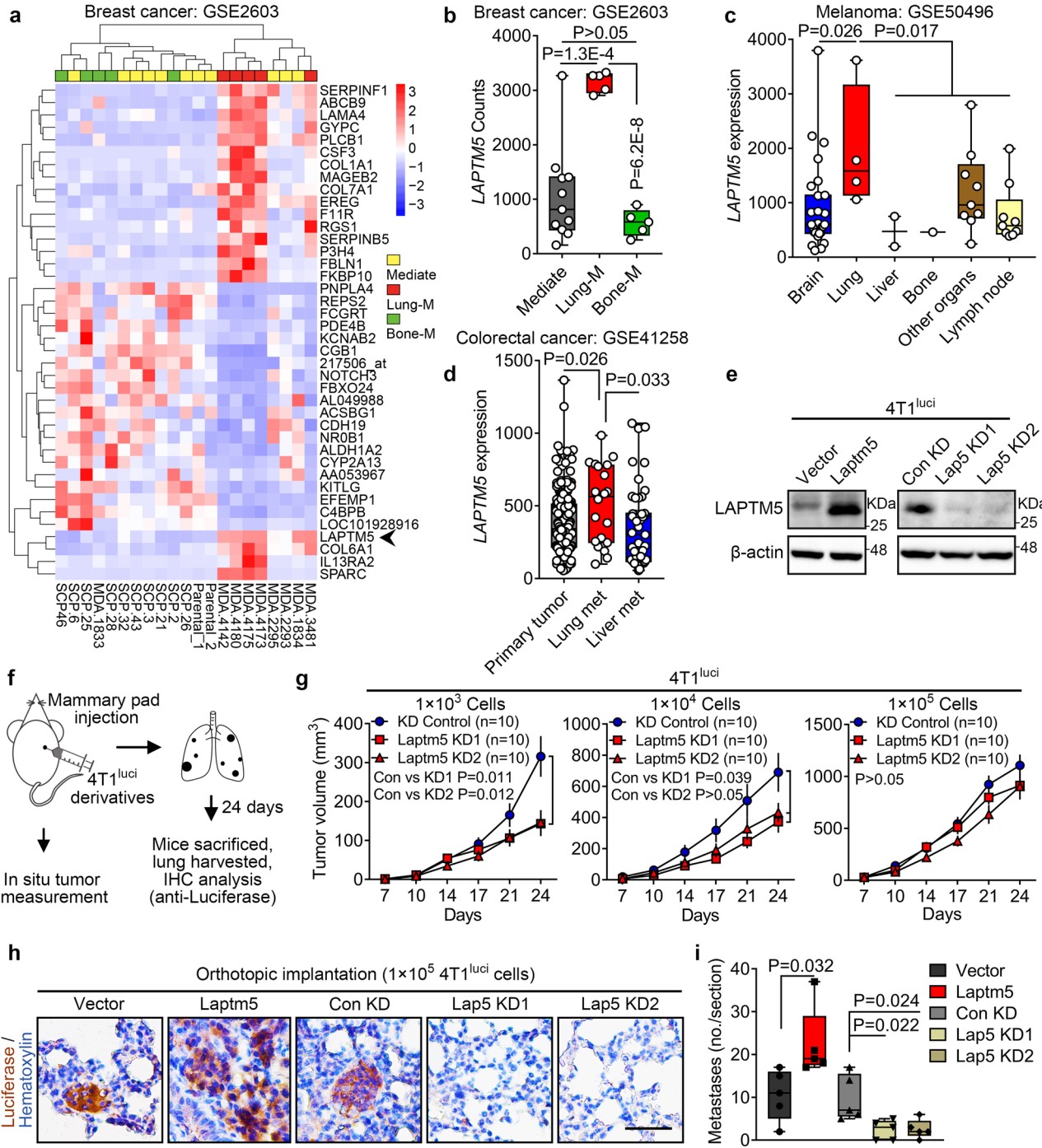

**Fig. 8 LAPTM5 is specifically activated in lung metastases of multiple cancers. a** Double clustering heatmap of top 20 up- and down-regulated genes in breast cancer (BCa) cells with lung metastatic activity (Lung-M, $n = 5$ cell lines) compared with populations with mediate activity (Mediate, $n = 11$ cell lines) and bone metastatic activity (Bone-M, $n = 5$ cell lines) in the GSE1206 dataset. **b** *LAPTM5* counts in BCa cells with mediate (Mediate, $n = 11$), lung metastatic (Lung-M, $n = 5$) and bone metastatic (Bone-M, $n = 5$) activity in the GSE1206 dataset. **c** *LAPTM5* expression in brain ($n = 27$ samples), lung ($n = 4$ samples), liver ($n = 2$ samples), bone ($n = 1$ sample), other organs ($n = 9$ samples) and lymph node ($n = 9$ samples) metastases from melanoma in the GSE50496 dataset. **d** *LAPTM5* expression in primary tumors ($n = 186$ samples), lung ($n = 20$ samples) and liver ($n = 47$ samples) metastases from colorectal cancer in the GSE41258 dataset. **e** IB analysis of control and Laptm5-overexpressing or control and Laptm5-silenced 4T1^luci cells. Immunoblots are representative of three biological replicates. **f** Schematic illustration for the mammary pad cell inoculation, in situ tumor measurement and lung metastases detection using the murine 4T1 breast cancer cell line. **g** Tumor volumes of control and Laptm5-silenced 4T1^luci cells inoculated in situ as described in (**f**). $n = 10$ mice per group. The data represent the mean ± SEM. **h** Representative IHC images of luciferase staining in lung sections harvested from mice treated as in (**f**). Scale bar, 50 μm. **i** Quantification of metastases per lung section at 24 days after mammary pad injection in each group ($n = 5$ mice per group). In **b**, **c**, **d**, and **i**, the data are presented as whisker plots: midline, median; box, 25–75th percentile; whisker, minimum to maximum values. Two-tailed Student's unpaired *t*-test was used for statistical analysis in all panels. Source data are provided as a Source data file.

For in vivo selection, a cell suspension containing $1 \times 10^4$ Renca[luci] in a volume of 100 μL phosphate buffer solution (PBS) was injected into the left ventricle (LV) of anesthetized 6 to 8-week-old male BALB/c mice. Similarly, a cell suspension of $1 \times 10^6$ 786O[luci/eGFP] was injected in the LV of 8-week-old male NOD/SCID mice. Mice were termly anesthetized and intraperitoneal injected with D-Luciferin (150 mg per kg body weight, 10 min prior to imaging) to monitor metastasis by in vivo imaging with the IVIS Spectrum Imaging System (Caliper Life Sciences, MA, USA). Only organs revealing signal of metastasis were harvested and further confirmed by ex vivo imaging. After that, tumor nodules from organs were aseptically dissected and digested in 0.25% EDTA containing trypsin, and inoculated in complete RPMI 1640 medium to acquire tumor cells from representative organs. After reaching confluency, derivative cells were purified and reinjected into mice for the next round of isolation.

To validate metastatic organotropism of derivative cells, cells (Renca[luci] and derivatives, $1 \times 10^4$ cells; 786O[luci/eGFP] and derivatives, $1 \times 10^6$ cells) were injected intracardially. To examine the effect of LAPTM5 on organ metastasis, cells (control and Laptm5-overexpressing Renca[luci], $1 \times 10^4$ cells; control and Laptm5-silenced Renca[LuM2b], $1 \times 10^4$ cells; and control and LAPTM5-overexpressing 786O[luci/eGFP], $1 \times 10^6$ cells) were injected intracardially. Luminescence was monitored every 5–6 days. The change in photon flux was determined using quantitative region-of-interest (ROI) analysis. Bone metastases were further examined with the X-Ray model of IVIS Spectrum Imaging System.

To examine lung colonization, cells (control and Laptm5-overexpressing Renca[luci], $1 \times 10^5$ cells; control and Laptm5-silenced Renca[LuM2b], $1 \times 10^5$ cells) were injected intravenously. Lungs were harvested at 1, 2, 7, 14, and 21 days after the injection. For the TET-inducible assays, Flag-Laptm5 (Tet-on) Renca[luci] cells ($1 \times 10^5$) were injected intravenously. Mice were administered with 2 mg/mL Dox in drinking water. Lungs were also harvested at 1, 2, 7, 14, and 21 days after injection. Lung metastatic lesions were confirmed by histological analysis.

For tumor initiation experiments, the indicated numbers ($1 \times 10^3$, $1 \times 10^4$, $1 \times 10^5$) of cells were suspended in a 1:1 mixture of PBS and growth-factor-reduced Matrigel, and subcutaneously implanted in buttocks or orthotopically implanted in the subcapsular of mice (Renca[luci] cells and derivatives), or injected into the mammary gland of 6 to 8-week-old female BALB/c mice (4T1[luci] cells and derivatives). Primary tumor growth was monitored twice a week by taking measurements of tumor length (L) and width (W). Tumor volume was calculated using the formula $LW^2/2$. For orthotopic settings, tumor formation and growth were assessed 21 days after implantation by in vivo imaging.

**RNA isolation, semi-qPCR, and real-time qPCR**. Total RNA was extracted with TRIzol reagent. cDNA synthesis of genes involved using PrimeScript RT Master Mix. Semi-PCR and real-time qPCR involved use of ChamQ Universal SYBR qPCR Master Mix with the StepOne Real-Time PCR System (Thermo Fisher Scientific, MA, USA). Experiments were carried out according to the manufacturer's instructions. Data were acquired with QuantStudio 6 Flex Software v 1.3. Products of semi-PCR were separated by agarose gel electrophoresis. The fold change in the gene expression was calculated using the comparative $Ct$ method, and two or three replicates were tested for each cDNA sample. *ACTB* or *Actb* were used as an internal reference. The sequences of the primers are listed in Supplementary Data 6.

**Library construction for RNA sequencing and sequencing procedures**. RNA was extracted from Renca[Parental], Renca[LuM2b], Renca[BrM2b], and Renca[BoM2] during the logarithmic growth phase. A total amount of 3 μg RNA per sample was used as input material for the RNA sample preparations. Sequencing libraries were generated using NEBNext® UltraTM RNA Library Prep Kit for Illumina® (NEB, MA, USA) according to the manufacturer's instructions and index codes were added to attribute sequences to each sample. In order to select cDNA fragments of preferentially 250–300 bp in length, the library fragments were purified with AMPure XP system (Beckman Coulter, Beverly, USA). Then, 3 μL USER Enzyme (NEB) was used with size-selected, adaptor-ligated cDNA at 37 °C for 15 min followed by 5 min at 95 °C before PCR. Then PCR was performed with Phusion High-Fidelity DNA polymerase, Universal PCR primers and Index (X) Primer. At last, PCR products were purified (AMPure XP system) and the quality of library was assessed on the Agilent Bioanalyzer 2100 system. Then, the index-coded samples were clustered on a cBot Cluster Generation System using TruSeq PE Cluster Kit v3-cBot-HS (Illumina, CA, USA) according to the manufacturer's instructions. After cluster generation, the library preparations were sequenced on an Illumina Novaseq platform and 125–150 bp paired-end reads were generated (NovoGene, Beijing, China).

**Immunoblotting (IB) and immunoprecipitation (IP)**. Total cell extracts were prepared using RIPA buffer containing a cocktail of proteinase inhibitors and a cocktail of phosphatase inhibitors. Then, proteins were denatured at 95 °C for 5 min and then separated by SDS-PAGE, then transferred to 0.22 μm PVDF membranes (Bio-Rad, Hercules, CA). After blocking with 5% skim milk, the membranes were incubated with specific primary antibodies overnight at 4 °C and secondary antibodies at room temperature (RT) for 1 h. Proteins were quantified using an electrochemiluminescence (ECL) system (Tanon, Shanghai, China). Data

were acquired using CLiNX platform and quantified using ImageJ 1.53 software. Polyclonal rabbit antibodies were raised against a peptide (PPKTPEGDPAPPPY-SEV) located near the C terminus of mouse LAPTM5[36].

For IP, after transfection or stimulation, cells were lysed in IP lysis buffer containing proteinase and phosphatase inhibitors. Specific primary antibodies were added to the lysates and incubated overnight at 4 °C, with homologous immunoglobulin G (IgG) as the control antibody. Then, Protein G Magnetic Beads (CST, MA, USA) were added in the system and incubated for 30 min at RT and then separated using the magnetic separation rack. Ultimately, conjoint proteins were eluted and sent for IB. Information on antibodies is provided in Supplementary Data 7.

**Immunofluorescence**. Briefly, cells were seeded on slides 24 h in advance. Then, culture medium was removed, and the cells were fixed with 4% paraformaldehyde (PFA) for 30 min. Cells were washed three times with PBS and permeabilized with 0.3% Triton X-100 for 15 min. Then cells were blocked with 5% BSA solution and incubated with specific primary antibodies overnight at 4 °C and fluorescence labelled secondary antibodies at RT for 1 h. Nuclei were stained with 0.2 mg/L DAPI for 5 min. Slides were observed and photographed using EVOS Auto 2 (Invitrogen) and EVOS FL Auto 2.0 Imaging System.

**TMT-labeled liquid chromatography-tandem mass spectrometry (LC-MS/MS)**. 786-O[LAPTM5] and Renca[Laptm5] cells treated or untreated with 10 ng/mL BMP4 for 60 min were used for IP assays with anti-Flag and rabbit IgG as control. After SDS-PAGE and Coomassie brilliant blue staining, DTT was added to the IP products until the final concentration was 100 mM. The IP products were then boiled for 5 min and cooled to RT. Next, 200 μL UA buffer was added and the products were then transferred to a 30 kD ultrafiltration tube, and centrifuged at $12,500 \times g$ for 15 min. The filtrate was discarded and the remnant was re-centrifuged. Then, 100 μL IAA buffer was added to the system (100 mM IAA in UA) and vortexed at 600 rpm for 1 min. The products were incubated in the dark at RT for 30 min, and centrifuged at $12,500 \times g$ for 15 min. After that, 100 μL UA buffer was added, followed by centrifugation at $12,500 \times g$ for 15 min; and repeated again. Next, 100 μL 0.1 M TEAB solution was added and centrifuged at $12,500 \times g$ for 15 min, repeated twice. Next, 40 μL Trypsin buffer (4 μg trypsin in 40 μL 0.1 M TEAB solution) was added and vortexed at 600 rpm for 1 min, then placed at 37 °C for 16–18 h. The products were transferred to a new collection tube and centrifuged at $12,500 \times g$ for 15 min. Then 20 μL 0.1 M TEAB solution was added and centrifuged at $12,500 \times g$ for 15 min, and the filtrate was collected. Later, C18 Cartridge was used to desalt and lyophilize the peptides, then 10 μL 0.1% formic acid was added to the peptides, and sent for TMT-labeled LC-MS/MS analysis (GeneChem, Shanghai, China). Data were collected with Thermo Q Exactive and software Proteome Discoverer 2.1.

**MTT assay**. Cultured cells in logarithmic phase were digested to obtain a cell suspension and seeded in 96-well plates (Renca and derivatives, $2 \times 10^3$ cells/well; 786-O and derivatives, $1 \times 10^4$ cells/well) and cultured in complete medium. After 12, 24, 48, 72 and 96 h, cells were incubated in 0.5 mg/mL MTT (3-4,5-dimethylthiazol-2-yl){-2,5-diphenyl tetrazolium bromide) for 2 h at 37 °C. MTT crystals were dissolved in DMSO and absorbance was measured at a wavelength of 570 nm using Tecan i-control system.

**Transwell invasion assay**. Diluted matrigel (BD Biosciences) was spread on the bottom of the upper transwell chamber (8 μm, Corning, NY, USA) and placed in 37 °C for 4 h to ensure matrigel condensation. The digested cells (Renca and derivatives, $1 \times 10^5$ cells/well; 786-O and derivatives, $2 \times 10^4$ cells/well) were seeded in the upper chamber to migrate, and incubated at 37 °C. After 24 h (Renca) or 10 h (786-O) incubation, chambers were fixed, then stained with the Crystal Violet Staining Solution (Beyotime), and counted under a light microscope.

**3D tumor sphere assay**. Single-cell suspensions (Renca, $4 \times 10^4$ cells/mL; 786-O, $1 \times 10^5$ cells/mL; 4T1, $4 \times 10^4$ cells/mL; MDA-MB-231, $1 \times 10^5$ cells/mL) were plated on ultra-low attachment plates (Corning) and cultured in serum-free RPMI 1640 or DMEM medium supplemented with 10 ng/mL epidermal growth factor (EGF) (Peprotech), 10 ng/mL fibroblast growth factor (FGF) (Peprotech) and 1 ng/mL B-27 (Peprotech) for 7 days. Tumor spheres were visualized by phase contrast microscope and photographed for counting.

**Clinical specimens**. Formalin-fixed RCC metastases from various organs (10 lung metastases, 22 bone metastases, 2 brain metastases), primary RCC tissues ($n = 150$), and noncancerous kidney tissues ($n = 106$) were collected from patients undergoing biopsies or surgical resection from 2005 to 2019. Cancerous tissue was classified according to the WHO classification. Patients' clinical and pathological information were collected and written informed consent was obtained from each patient. Ethics approval was obtained from the Nanjing University Medical School affiliated Nanjing Drum Tower Hospital.

**Hematein-eosin (H&E) and immunohistochemical (IHC) staining**. Paraformaldehyde fixed tissues were cut into 3 μm slices and attached to a highly adherent slide. Next, slides were placed at 75 ℃ for 2 h, then dewaxed in xylene for 3 min thrice, and placed in 100, 90, 80, and 70% ethanol solution 2 min each to re-hydrate. For H&E staining, re-hydrated slides were stained in hematoxylin for 1 min and eosin for 1 min. For IHC staining, re-hydrated slides were blocked with 5% BSA solution and then incubated with specific primary antibodies overnight at 4 ℃ and HRP-linked secondary antibodies (OriGene, Beijing, China) at RT for 1 h. Finally, the detection was performed with DAB detection kit (ZsBio, Beijing, China). Data were acquired using Leica Microsystems and Leica Application Suite v 4.12.0. Multiple random fields on the slice were chosen for analyzing, staining intensity was scored as 0 (negative), 1 (low), 2 (moderate), 3 (high), and 4 (extremely high). Staining range was scored as percentage of each staining intensity. The final score was defined as the sum of product of the intensity scores and staining range; a score ≤ 160 was defined as low expression and > 160 high expression in clinical samples.

**Bioinformatics analysis**. Array data for Jon_Renal_Cancer were obtained from the supplementary data of Jone's study (PMID: 16115910), while processed matrix array data for GSE2603, GSE50496, and GSE41258 were collected from Gene Expression Omnibus (GEO) database. GSEA was analyzed by pre-Ranked GSEA (preRankedGSEA) on genes ranked by Spearman correlation coefficient. Briefly, the spearman correlation coefficients between LAPTM5 and all other genes were calculated using TCGA KIRC or KIRP dataset, separately. The correlation coefficients were sorted in descending order and submitted to "GSEA" function of R "clusterProfiler" package with default parameters. The gene sets were obtained from the Molecular Signatures Database (MSigDB 7.1) and the pathways curated from the Pathway Interaction Database (PID) were used in this analysis. The GSEA results were further visualized by R "enrichplot" package.

**Statistical analysis**. The data presentation and statistical analyses are described in the figure legends. $\chi^2$ test and univariate and multivariate analyses were performed with IBM SPSS Statistics 21 software. The remaining data analyses were performed by Graphpad Prism 7 software. Differences with P values < 0.05 were considered statistically significant.

**Reporting summary**. Further information on research design is available in the Nature Research Reporting Summary linked to this article.

## Data availability

The RNA-seq data generated in this study have been deposited in the National Center for Biotechnology Information Sequence Read Archive (SRA) database under accession code PRJNA724865. Processed data of breast cancer (accession number: GSE2603;), melanoma (accession number: GSE50496;) and colorectal cancer (accession number: GSE41258;) were collected from Gene Expression Omnibus (GEO) database. Processed data of Jon_Renal_Cancer were collected from supplementary information of this paper (https://doi.org/10.1158/1078-0432.CCR-04-2225)[30]. All the other data supporting the findings of this study are available within the article and its Supplementary Information files. A reporting summary for this article is available as a Supplementary Information file. Source data are provided with this paper.

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

## Acknowledgements

This work was supported by grants from the National Natural Science Foundation of China (82002681, 81772710, 81972388, 82070703, 81972387 and 21877060), the Natural Science Foundation of Jiangsu Province (BK20200123) and the Project of Invigorating Health Care through Science, Technology and Education Jiangsu Provincial Key Medical Discipline (ZDXKB2016014). We thank Vazyme Biotech for the production of polyclonal antibodies against mouse LAPTM5.

## Author contributions

H.G., B.J., and X.Z. devised and coordinated the project. B.J. performed all the experiments with help from W.C. (the third author), W.D., M.D., WM.C., H.Q., W.C. (the ninth author), J.G., M.C., K.H., T.L., and Y.D. B.L. performed bioinformatics analyses. B.J., W.D., and K.H. performed experiments with the Renca metastasis model. Y.F. collected clinical samples and performed IHC analysis. B.J., C.Y., and H.G. wrote the manuscript. All authors revised the manuscript.

## Competing interests

The authors declare no competing interests.
