## [Peer Review File · Nature Communications]

Lysosomal protein transmembrane 5 promotes lung-specific metastasis by regulating BMPR1A lysosomal degradationREVIEWER COMMENTS

Reviewer #1 (Remarks to the Author):

This manuscript reports on the identification of the lysosomal protein transmembrane 5 (LAPTM5) as a key molecule in the process that promotes lung-specific metastasis of renal cell carcinoma. The authors show that upregulation of LAPTM5 supports the self-renewal capacity and stemness of renal cancer cells. They further demonstrate that cancer cells metastasizing to the lungs upregulate LAPTM5 to recruit the ubiquitin ligase WWP2 that in turn mediates the lysosomal sorting, ubiquitination and degradation of the bone morphogenic protein receptor BMPR1A. Abnormal degradation of BMPR1A impairs the inhibitory signals of lung-stroma- derived BMP, enabling metastasis-initiating cells to maintain malignant stem-like features and to spread/grow. Although the focus of this paper was on metastatic renal cell carcinoma, the authors show that this novel LAPTM5-mediated degradation pathway occurs in other types of carcinomas that metastasize to the lungs and propose LAPTM5 as a potential therapeutic target.

Overall, this is a well written manuscript. The data are solid and presented following a consequential and logic explanation of the underlying hypothesis. The authors use an appropriate set of experimental approaches and controls to answer their questions. The figures are of quality and clear, but occasionally far too small. The conceptual framework is innovative in that it led to the identification of a post translationally regulated mechanism promoting metastases of renal cancer cells to the lung, as well as of key mediators of this process.

There are a few points of concern that require clarifications or corrections before final recommendation for publication can be made:

General remarks:

Overall, it is not clear how the statistical analyses were done, given that statistics are included only for some of the experiments. In addition, it is not well specified how many experiments and replicates have been used throughout. In some of the graphs (for example Fig. 2c) it is unclear whether the data refer to separate independent experiments or duplicates of the same assays. No statistical analysis is shown and no reference to the number of samples is given in the figure legend.

The authors should at least discuss whether they also detected an increase in the autophagy pathway in metastatic renal cancer cells that could occur in parallel to the enhanced endocytosis of BMPR1A and its lysosomal degradation.

Line 85 – The authors state that different organ derivatives showed distinct cell morphologies. (Figure S1C). However, this cannot be deduced from the displayed images that are too small and lack details when zoomed in. How does the cell morphology of the metastatic tumors in tissue sections compare to that of these organ derivatives?

Line 118 – I would suggest rephrasing the sentence: “analysis of these two datasets identified three genes... that appear to be activated specifically in ...”. Looking at Table S2, the three genes the authors refer to were not the only one identified through this analysis that appeared upregulated in RCC derived from lung metastasis.

Line 126 – The authors state: “However, two other mediators CTSS and IGFBP5, were not as highly or reproducibly upregulated in lung derivatives.” This is misleading as stated because at this point, they did not have any indications that these genes were mediators of metastases. Moreover, how do the authors determined that the expression of these genes was not as highly or reproducibly upregulated if no statistics are included in Figures S3A and S3B? In fact, IGFBP5 seems upregulated. Please rephrase and/or add statistical analysis. Please specify the number of experimental replicates in the figure legend.

Line 129 – It is very difficult to evaluate the IHC results shown in Figure 2E. Please consider adding higher magnification images or insets that will give a clearer display of the staining's.

Line 153 – This paragraph would be easier to follow if the order of the panels in Figure 3 reflects the way the data are discussed in the text.

The authors state that LAPTM5 expression in RCC cells did not affect the subcutaneous tumor growth rate, but this is not what is shown in the Figures 3H and 3I. The overexpression or knockdown of LAPTM5 in Renca cells appears to affect tumor volume when 1×10^3 cells were injected. Please clarify this point.

Line 174 to 178 – This paragraph refers to results that have been already discussed in Line 152. Here the conclusion was that Laptm5 overexpression significantly enhanced the subcutaneous tumor initiation ability of Renca cells. This contrasts with what the authors stated earlier (in line 152), namely that Laptm5 overexpression did not affect subcutaneous tumor growth rate. In both cases Figures 3H and 3I are referenced. Please correct these opposing statements.

Line 179 – Here the authors refer to the expression levels of other ESC transcription factors that appear elevated in LAPTM5 overexpressed 786-O and Renca cells; were these transcription factors (NANOG, OCT4, SOX2 and KLF4) downregulated in LAPTM5 knockdown cells?

Line 190 – By looking at Tables S3-S4 it does not seem that BMP signaling was the only pathway that exhibited a negative correlation with LAPTM5 levels, as the authors stated. Please clarify.

Line 197 – There is no quantification or statistical analysis done in Figures 4D and S5B.

Line 203 – 204 – In reference to Figure 4F, LuM2bCon KD Renca cells which have endogenous increased levels of LAPTM5 should respond to BMP- induced Smad 1/5/8 phosphorylation similarly to RencaLAPTM5, but they do not, since they behave as the RencaVector. Please clarify this point.

Line 207 – It would be appropriate if the experiments performed in 786-O cells were also done in the RencaLAPTM5 and LuM2bLap5 KD cells. The authors should consider including these experiments.

Line 212 – Which transcription factors in Renca or 786-O cells were inhibited by BMP4 and reactivated by LAPTM5? Please specify. For example, Klf4 was not. Please indicate the number of experimental replicates for Figure 4I.

Line 230 – 233 – Would the BMPR1A levels increased in LuM2bLap5 KD cells? This could be an important control to add.

Line 258 – Here the authors did not use the Renca cells anymore but switched to 786-O and even 293T cells, without any explanation. Please clarify their choice of using different cells.

Line 295 – The authors say that BMPR1A ubiquitination was abolished by the expression of WWP2C838A mutant, however this is not what is shown in Figure 6B, where WT and C838A mutant have both ubiquitination of BMPR1A at apparently similar levels. Please clarify and correct this point.

Line 298 – “significantly reduced BMPR1A ubiquitination”. There is no statistical analysis done to know that it is a significant change.

Line 362 – Again there was a change of cells used for this experiment with no explanation. Please clarify.

Table S2 – Please add p value and adj. p value.

Figures 2C and 2D – how many experiments and replicates were used? Please provide statistical analysis.

Figures S3C and S2D – please add the number of experimental replicates Was the p value calculated on a duplicate of 1 experiment or on several experiments?

Figure S4C – p value is missing.

Figure 3K; 4K-4M; S5I – again please specify the number of replicates. It looks like that statistical analysis was done using only 2 values, which is not robust enough. Please clarify this point.

Reviewer #2 (Remarks to the Author):

This manuscript investigated the role of LAPT5 on lung-specific metastasis of RCC. The authors isolated and characterized organ-specific metastatic derivatives of two RCC cell lines and identified LAPT5 as a key mediator of lung-specific metastasis of RCC. In addition, the authors suggested that LAPT5 promotes self-renewal of RCC cells in the lung by attenuation (or inhibition) of BMP-induced phosphorylation of smad1/5/8. Finally, the authors suggested that LAPT5-mediated inhibition of BMP-pathway is due to degradation of BMPR1A via interaction among LAPT5-WWP2-BMPR1A.

While this manuscript provides some information on the role of LAPT5 in the lung metastasis of RCC, this manuscript suffers from several weaknesses. The specific comments are as follows.

Major comments.

1. Fig 3: The involvement of LAPT5 in the inhibition of metastasis initiation needs to be investigated more vigorously. The authors performed IV injection of LAPT5 OE or KD cell, followed by IHC. However, to further support their claims on the specific role of LAPT5 in promotion of the micro-metastasis formation, the authors should perform similar experiments with TET-inducible ShLAPT5 or LAPT5.
2. Fig3H and I: The role of LAPT5 in tumor initiation should be tested in the orthotopic setting instead of subcutaneous injection by using renal subcapsular injection. And the staining of BMPR1 should be accompanied.
3. Fig 4A-C. the authors stated that “We found that BMP signaling was the only pathway that exhibited a significantly negative correlation with LAPT5 levels in both KIRC and KIRP patients” and focused on the LAPT5 and BMP pathway thereon. This reviewer could see that the reason the authors focused in the BMP pathway. However, it is equally possible that LAPT5 can promote lung-specific metastasis by regulating the pathways showing a positive correlation. Indeed, in Table S3,4, there are several pathways that show good positive correlation with LAPT5. Therefore, the authors should provide a clear rationale for initially focusing on the BMP pathway in addition to negative correlation and bioactive BMPs in the lung.
4. Authors have shown, in Fig3F, that KD or OE of LAMPT5 reduced and increased sphere formation, respectively. According to Fig 4K, however, the BMPR1 protein levels are similar between control and LAMPT5-manipulated cells in the absence of exogenous BMP4. If LAMPT5 - mediated BMPRA1 degradation is a key pathway for LAMPT5-induced sphere formation, how the results in Fig 3F can be explained in which presumably BMPR1 protein levels are similar between cells used in the experiment? The authors should collect spheres from these experiments and examined for the BMPR1 protein levels.
5. Fig 4G should be repeated with LAMPT5 KD-BMPR1A co-knockdown cells and/or LAMPT5 KD-DMH1 treatment.
6. The authors have shown the lung metastasis-promoting roles of LAMPT5. However, this study completely lacks in vivo experiments supporting whether this is via LAMPT5KD-WWA-BMPR1A axis. The authors should perform in vivo experiments to clearly showing this.
7. Fig5E-H, the authors performed several biochemical experiments to show that LAMPT5-WWA-BMPR1A forms a complex which leads to degradation of BMPR1A. However, a majority of experiments were performed with cells overexpressing LAMPT5-WWA-BMPR1A. Thus, similar experiments should be performed with endogenous proteins.
8. Fig 5E-H, the authors should perform serial IP to clearly show LAMPT5-WWA- BMPR1 are indeed in the same complex.
9. Fig 6, the experiments were performed with 293T cells only. The similar experiments should be performed in RCCs.
10. According to a previous paper “Functional Proteomics Mapping of a Human Signaling Pathway” (Genome Res, 2004), this paper showed that LAPT5 interacts with Smurf2 and activated the TGFb pathway but not BMP pathway. The author should cite this paper and discuss the potential

explanation for the differences between this paper and the current manuscript in the discussion. Also, the paper showed TGF β treatment increases LAPMT5 expression. This may provide a clue on why LAPMT5 is overexpressed in LM derivatives. The authors may want to test this in their experimental system.

Minor comments.

1. High resolution cell morphology S1c is needed
2. Does LAPMT4 KD revert the morphology of the RCC LM cell back to the par?
3. It would be better to provide a bit more information on LAPMT5 on page 6 where it was first mentioned in the manuscript.
4. The figure numbers in the text should be according to the order of figures. (e.g pp7 Fig 3H and J come first than 3A.)

Reviewer #3 (Remarks to the Author):

This is a thorough and solid study introducing the role of LAPTM5 to sustain self-renewal and confer lung-tropism to RCC tumor cells. It was a pleasant experience to review this manuscript. Topic/hypothesis is of interest and significance. Logic flow is smooth and easy to follow. Experiment design and data presentation is clean and neat. Yet the reviewer founds several conceptual caveats that need to be addressed before this manuscript is accepted for publication in Nature Communication.

The authors tend to attribute RCC's lung tropism to the self-renewal program induced by LAPTM5. Reasoning that self-renewal/stemness per se is not an organ-specific phenotype, the authors' hypothesis may face more challenges and concerns, compared to other peer studies that focused on more organ-specific pathways.

1, one fundamental premise of the authors' hypothesis is the exceptional enrichment of BMPs in lung, which should suppress the activity of ESC transcription factors in all but the lung-tropic cells. However, the uneven expression of BMP in different organs is not well-evident in vivo. No data or reference were provided to support this notion except for a simple annotation at Line 192 [...bioactive BMPs (deficient in the bone and brain) from lung stroma...]. This is a major caveat.

2, conceptually, "organotropism" needs to be interpreted from two aspects: a. Why only lung-derivatives can thrive in lung; b. Why the other organs are not favorable to lung-tropic cells? Question b is not discussed in this manuscript. Indeed, in most other peer studies focusing on organ-specific pathways, question b is usually self-evident (the indicated gene/pathway is not activated or even not present in other organs). However, this study is different from them. The authors are recommended to at least discuss why the self-renewal phenotype in lung-derivatives is less robust in brain and bone. Another relevant question is, can LAPTM5 overexpression in bone- or brain- derivatives (not the parental cells) convert their organ tropism?

3, At least in some experiments (like sphere assay in Fig. 3F), the function of LAPTM5 appears BMP-independent. The authors are recommended to clarify it.

4, It is a bit regretted that no pharmacological assay was performed for pre-clinical trial. Otherwise the impact of this paper will be considerably increased.

Minor comments:

- 1, some "non-significant" results in Fig.3P are likely due to poor data performance (large deviation/outlier) instead of biological reason.
- 2, Fig. 7H, LUNG-metastasis-free survival curve should be more relevant. Ideally, the LAPTM5 expression is expected to dictate ONLY lung-metastasis-free survival.

Point-to-point response to reviewers:

NCOMMS-21-27982-T

Lysosomal Protein Transmembrane 5 Promotes Lung-Specific Metastasis by Regulating BMPR1A Lysosomal Degradation**Reviewer #1:**

This manuscript reports on the identification of the lysosomal protein transmembrane 5 (LAPTM5) as a key molecule in the process that promotes lung-specific metastasis of renal cell carcinoma. The authors show that upregulation of LAPTM5 supports the self-renewal capacity and stemness of renal cancer cells. They further demonstrate that cancer cells metastasizing to the lungs upregulate LAPTM5 to recruit the ubiquitin ligase WWP2 that in turn mediates the lysosomal sorting, ubiquitination and degradation of the bone morphogenic protein receptor BMPR1A. Abnormal degradation of BMPR1A impairs the inhibitory signals of lung-stroma-derived BMP, enabling metastasis-initiating cells to maintain malignant stem-like features and to spread/grow. Although the focus of this paper was on metastatic renal cell carcinoma, the authors show that this novel LAPTM5-mediated degradation pathway occurs in other types of carcinomas that metastasize to the lungs and propose LAPTM5 as a potential therapeutic target.

Overall, this is a well written manuscript. The data are solid and presented following a consequential and logic explanation of the underlying hypothesis. The authors use an appropriate set of experimental approaches and controls to answer their questions. The figures are of quality and clear, but occasionally far too small. The conceptual framework is innovative in that it led to the identification of a post translationally regulated mechanism promoting metastases of renal cancer cells to the lung, as well as of key mediators of this process.

There are a few points of concern that require clarifications or corrections before final recommendation for publication can be made:

General remarks:

1. Overall, it is not clear how the statistical analyses were done, given that statistics are included only for some of the experiments. In addition, it is not well specified how many experiments and replicates have been used throughout. In some of the graphs (for example Fig. 2c) it is unclear whether the data refer to separate independent experiments or duplicates of the same assays. No statistical analysis is shown and no reference to the number of samples is given in the figure legend.

Reply: We thank the reviewer for these suggestions. As requested, we have provided the details of statistical analysis and the duplicates of assays in all figure legends in the revised manuscript. It should be noted that we changed some of the statistical tests to more appropriate statistical methods in some figures (for example, two-tailed Student's *t*-test to two-

way ANOVA test in Fig.S4A-C), which gave a different p-value from the previous version. These changes are highlighted in the revised manuscript

- The authors should at least discuss whether they also detected an increase in the autophagy pathway in metastatic renal cancer cells that could occur in parallel to the enhanced endocytosis of BMPR1A and its lysosomal degradation.

Reply: We understand the rational for this question and agree with the reviewer. We have carried out additional experiments to evaluate the activation of autophagy pathway. No obvious changes in LC3B-I/II levels (a marker of autophagy) were detected in lung-met derivatives or LAPTM5-overexpressing 786-O or Renca cells, suggesting autophagy is not involved in BMPR1A degradation process.

To gain more insights into the pathways involved in BMPR1A degradation, we performed additional IF experiments and showed an increase in the late endosome and/or lysosome formed by LAPTM5, in parallel to the enhanced endocytosis of BMPR1A in lung metastatic RCC cell (786O^{LuM1a}). Moreover, BMP4 treatment induced a more dramatic increase in BMPR1A endocytosis. These new results are added to Figure S7K.

- Line 85 – The authors state that different organ derivatives showed distinct cell morphologies. (Figure S1C). However, this cannot be deduced from

the displayed images that are too small and lack details when zoomed in. How does the cell morphology of the metastatic tumors in tissue sections compare to that of these organ derivatives?

Reply: We apologize for the quality of these images. To help readers to appreciate the differences in cell morphology, we have added representative brightfield images of cells to this figure (see below). Compared with parental cells, the metastatic derivatives exhibited a more multilegged and variform form in LuM2b, a larger form in BoM2, and a more compact form in BrM2b, the underlying cause of changes in cell morphology is not the focus of this project. In tissue sections, we also observed larger cells in bone met and more compact cells in brain met, cell morphology in lung met is indistinguishable from brain met.

4. Line 118 – I would suggest rephrasing the sentence: “analysis of these two datasets identified three genes... that appear to be activated specifically in ...”. Looking at Table S2, the three genes the authors refer to were not the only one identified through this analysis that appeared upregulated in RCC derived from lung metastasis.

Reply: We thank the reviewer for pointing this out. Table S2 showed the integrated analysis result for Figure 2B, indeed *CTSS*, *LAPTM5* and *IGFBP5* were the only three genes that were upregulated both in our dataset and the Jon_Renal_Cancer dataset. We have rephrased the sentence to: “Integrated analysis of these two datasets identified three genes: cathepsin S (*CTSS*), lysosomal protein transmembrane 5 (*LAPTM5*), and insulin-like growth factor binding protein 5 (*IGFBP5*), that appear to be activated specifically in RCC cells from lung metastases in

both datasets.”

5. Line 126 – The authors state: “However, two other mediators CTSS and IGFBP5, were not as highly or reproducibly upregulated in lung derivatives.” This is misleading as stated because at this point, they did not have any indications that these genes were mediators of metastases. Moreover, how do the authors determined that the expression of these genes was not as highly or reproducibly upregulated if no statistics are included in Figures S3A and S3B? In fact, IGFBP5 seems upregulated. Please rephrase and/or add statistical analysis. Please specify the number of experimental replicates in the figure legend.

Reply: We apologize for not stating this result clearly. We have reorganized the results of Fig. 2C, Fig. S3A and S3B, and displayed it in the new Fig. 2C (see figure below). We think it would be more intuitive to show the changes in *Ctss*, *Laptm5* and *Igfbp5* mRNA expression levels in all organ derivatives as normalized to the parental cell. Meanwhile, we rephrased the statement to “Among these three genes, LAPTM5 showed much more prominent elevation than CTSS and IGFBP5 in lung derivatives.”

6. Line 129 – It is very difficult to evaluate the IHC results shown in Figure 2E. Please consider adding higher magnification images or insets that will give a clearer display of the staining’s.

Reply: We have replaced the IHC images in Figure 2E with higher magnification, we also added an inset image of HE staining to show the site of bone metastasis.

7. Line 153 – This paragraph would be easier to follow if the order of the

panels in Figure 3 reflects the way the data are discussed in the text.

Reply: We thank the reviewer for pointing this out. We have deleted the callout of 3H and 3I here “or the subcutaneous tumor growth rate (Figures 3H and 3I)”, which should not affect the logic of this section.

8. The authors state that LAPTM5 expression in RCC cells did not affect the subcutaneous tumor growth rate, but this is not what is shown in the Figures 3H and 3I. The overexpression or knockdown of LAPTM5 in Renca cells appears to affect tumor volume when 1×10^3 cells were injected. Please clarify this point.

Reply: In Figures 3H and 3I, we performed the classical *in vivo* tumor-initiation assay, i.e., using serial dilution of tumor cells for sub-cu tumor inoculation. Only tumor-initiating cells have the ability to support tumor growth when limited numbers of cells are implanted [1-3]. Therefore, the results from 3H and 3I were interpreted as the effect of LAPTM5 on tumor-initiation but not on tumor growth rate, especially when only 1×10^3 cells were implanted. The growth rate between control cells and cells with LAPTM5 overexpression or KD were not different when 1×10^5 cells were implanted, which is still less than but closer to the “normal” amount of cells used in a sub-cu growth rate test.

9. Line 174 to 178 – This paragraph refers to results that have been already discussed in Line 152. Here the conclusion was that Laptm5 overexpression significantly enhanced the subcutaneous tumor initiation ability of Renca cells. This contrasts with what the authors stated earlier (in line 152), namely that Laptm5 overexpression did not affect subcutaneous tumor growth rate. In both cases Figures 3H and 3I are referenced. Please correct these opposing statements.

Reply: Please refer to the answers to the above two questions.

10. Line 179 – Here the authors refer to the expression levels of other ESC transcription factors that appear elevated in LAPTM5 overexpressed 786-O and Renca cells; were these transcription factors (NANOG, OCT4, SOX2 and KLF4) downregulated in LAPTM5 knockdown cells?

Reply: Good question. We have performed these experiments. As expected, the mRNA levels of *Nanog*, *Oct4* and *Sox2* but not *Klf4* were downregulated.

11. Line 190 – By looking at Tables S3-S4 it does not seem that BMP signaling was the only pathway that exhibited a negative correlation with LAPT5 levels, as the authors stated. Please clarify.

Reply: We apologize for not stating this result clearly. Since the majority of LAPT5 literature report its negative regulation of membrane receptors, we emphasized on signaling pathways that are suppressed by LAPT5. From Table S3, there were 2 pathways that exhibited a negative correlation with LAPT5 levels in the KIRC dataset; from Table S4, 14 pathways exhibited negative correlation with LAPT5 levels in the KIRP dataset. Out of the 2 and 14 pathways, BMP signaling turned out to be the only pathway that is overlapped in both the KIRC and the KIRP cohort. Therefore, we made the statement that “We found that BMP signaling was the only pathway that exhibited a significantly negative correlation with LAPT5 levels in both KIRC and KIRP patients.”

12. Line 197 – There is no quantification or statistical analysis done in Figures 4D and S5B.

Reply: Fig. S5B was initially presented as the quantification analysis of Fig. 4D. We reasoned that for western blots like the ones presented in Fig. 4D, a detailed quantification and statistical comparison among all groups is both unrealistic to perform and provides little additional information meaningful to the understanding of this data. We have therefore removed Fig. S5B from the revised manuscript to avoid reader confusion.

13. Line 203 – 204 – In reference to Figure 4F, LuM2b^{Con KD} Renca cells which have endogenous increased levels of LAPT5 should respond to BMP-induced Smad 1/5/8 phosphorylation similarly to Renca^{LAPT5}, but they do not, since they behave as the Renca^{Vector}. Please clarify this point.

Reply: Good question. We have repeated the experiment in Fig. 4F several times and got similar results. Firstly, in this experiment, all cells respond to BMP4 treatment, with an initial increase in Smad 1/5/8 phosphorylation followed by gradual diminution; the higher expression of LAPT5 could either delay this response (Renca^{Laptm5} v.s. Renca^{Vector}) or reduce the degree of response (LuM2b^{Con KD} v.s. LuM2b^{Laptm5 KD}).

Secondly, the results of the three panels in Fig. 4F came from three independent WB experiments, which might not be comparable to each other. Thirdly, the endogenous levels of LAPTM5 in LuM2b^{Con KD} is not as high as ectopically overexpressed LAPTM5 in Renca^{Lap5} cells, which may partly explain the discrepancies pointed out by the reviewer.

14. Line 207 – It would be appropriate if the experiments performed in 786-O cells were also done in the Renca^{LAPTM5} and LuM2b^{Lap5 KD} cells. The authors should consider including these experiments.

Reply: We have repeated these experiments in Renca cells and add the results (see below) to new Fig. S6D.

15. Line 212 – Which transcription factors in Renca or 786-O cells were inhibited by BMP4 and reactivated by LAPTM5? Please specify. For example, Klf4 was not. Please indicate the number of experimental replicates for Figure 4I.

Reply: We apologize for not stating this result clearly. We have specified the transcription factors in the revised manuscript and indicated the number of experimental replicates in the figure legend.

16. Line 230 – 233 – Would the BMPR1A levels increased in LuM2b^{Lap5 KD} cells? This could be an important control to add.

Reply: Good question. We detected an increase in BMPR1A levels in LuM2b^{Lap5 KD} cells compared with LuM2b^{Con KD} cells, and have added this data to new Fig. S6J.

17. Line 258 – Here the authors did not use the Renca cells anymore but switched to 786-O and even 293T cells, without any explanation. Please

clarify their choice of using different cells.

Reply: Good question. As the reviewer pointed out, three major cell model systems were used in our manuscript: the murine Renca cell line, the human 786-O cell line and the 293T tool cell line. Several factors were considered when choosing the most appropriate cell line for each experiment. Firstly, the establishment of lung met derivatives were best performed with the murine Renca cell line in the BALB/c immune-competent mouse background. Therefore, the first half of the work were mainly carried out in the Renca system. However, we also managed to generate 786-O derivatives in NOD/SCID mice with severe immune deficiency and supplemented these data as additional evidence. Secondly, there were no commercially available antibodies against murine LAPTM5, although we have generated this antibody in-house, we only verified its applications in WB and IHC assays, therefore, we performed subsequent mechanism studies using the human 786-O system as the primary cell model, supplemented with Renca data. Thirdly, the embryonic kidney cell line 293T is a classical tool cell line widely used for studying protein-protein interaction in cells because it can be easily transfected with multiple plasmids overexpressing tagged proteins [4-6], that's why we used this cell line for most of the protein-protein interaction experiments. In the revised paper, we have also supplemented the IP results in 293T results with new data in 786-O (shown below and in new Fig. 6A-D).

18. Line 295 – The authors say that BMPR1A ubiquitination was abolished by the expression of WWP2^{C838A} mutant, however this is not what is shown in Figure 6B, where WT and C838A mutant have both ubiquitination of BMPR1A at apparently similar levels. Please clarify and correct this point.

Reply: We thank the reviewer for pointing this out and apologize for the overstatement. The original blot shown in Fig. 6B is overexposed to show the difference in other lanes. We have quantified the signal in each lane and there is indeed a reduction but not “abolition” in BMPR1A ubiquitination with C838A mutation. Besides, we have repeated this experiment in 786-O cells and observed similar but more potent results (new Figure 6B). We have rephrased the text in the revised manuscript.

19. Line 298 – “significantly reduced BMPR1A ubiquitination”. There is no statistical analysis done to know that it is a significant change.

Reply: We have removed the word “significantly” in the revised

manuscript.

20. Line 362 – Again there was a change of cells used for this experiment with no explanation. Please clarify.

Reply: Please refer to the answer to question 17.

21. Table S2 – Please add p value and adj. p value.

Reply: We have added the p value and adj. p value.

22. Figures 2C and 2D – how many experiments and replicates were used? Please provide statistical analysis.

Reply: We have rearranged Figures 2C and 2D to facilitate the understanding of the data and added necessary information to the figure legend.

23. Figures S3C and S2D – please add the number of experimental replicates Was the p value calculated on a duplicate of 1 experiment or on several experiments?

Reply: Figure S2D is a volcano plot. If the reviewer is referring to Figures S3C and S3D, the p value was calculated on three experiments; we have added the details in the figure legend.

24. Figure S4C – p value is missing.

Reply: We have added the p value in the figure.

25. Figure 3K; 4K-4M; S5I – again please specify the number of replicates. It looks like that statistical analysis was done using only 2 values, which is not robust enough. Please clarify this point.

Reply: We thank the reviewer for pointing this out. In Figures 3J and 3K, statistical analysis of SOX2 was done using only 2 values, so did data in Figure 4L. We have repeated these experiments and replaced these figures with new results and statistics in the revised manuscript. Statistical analysis in Figures new Fig. 4M and S6K were done using 3 values.

Reviewer #2:

This manuscript investigated the role of LAPTM5 on lung-specific metastasis of RCC.

The authors isolated and characterized organ-specific metastatic derivatives of two RCC cell lines and identified LAPTM5 as a key mediator of lung-specific metastasis of RCC. In addition, the authors suggested that LAPTM5 promotes self-renewal of RCC cells in the lung by attenuation (or inhibition) of BMP-induced phosphorylation of smad1/5/8. Finally, the authors suggested that LAPTM5-mediated inhibition of BMP-pathway is due to degradation of BMPRI1A via interaction among LAPTM5-WWP2-BMPRI1A.

While this manuscript provides some information on the role of LAPMT5

in the lung metastasis of RCC, this manuscript suffers from several weaknesses.

Major comments:

1. Fig 3: The involvement of LAPTM5 in the inhibition of metastasis initiation needs to be investigated more vigorously. The authors performed IV injection of LAPTM5 OE or KD cell, followed by IHC. However, to further support their claims on the specific role of LAPTM5 in promotion of the micro-metastasis formation, the authors should perform similar experiments with TET-inducible ShLAPTM5 or LAPTM5.

Reply: We thank the reviewer for this suggestion. We have performed new experiments using the TET-on system and obtained similar results (see below). We have added these data as new Figs. S4K-S4P.

Method: For the TET-inducible assays, Renca^{luci} cells were infected with TET-inducible Flag-Laptm5/Control lentivirus and selected with puromycin for 7 days. After that, cells were treated with doxycycline (Dox, 1 µg/mL or 2 µg /mL) for 48 h and then total cell lysate was harvested for IB detection. To examine lung colonization, Flag-Laptm5 (Tet-on) Renca^{luci} cells (1×10^5) were injected intravenously. Afterwards, mice were administered with 2 mg/mL Dox in their drinking water. Lungs were harvested at 1, 2, 7, 14, and 21 days after injection. Lung metastatic lesions were confirmed by histological analysis.

2. Fig3H and I: The role of LAPT5 in tumor initiation should be tested in the orthotopic setting instead of subcutaneous injection by using renal subcapsular injection. And the staining of BMPR1A should be accompanied.

Reply: We thank the reviewer for this suggestion. We have performed new experiments in the orthotopic setting and obtained similar results (see below). We have added these data as new Figures S5B and S5C.

Method: For tumor initiation experiments in the orthotopic setting, the indicated numbers (1×10^3 , 1×10^4 , 1×10^5) of cells were suspended in a 1:1 mixture of PBS and growth-factor-reduced Matrigel, and inoculated by

renal subcapsular injection (Renca^{luci} cells and derivatives). Tumor formation and growth were monitored 21 days after implantation by *in vivo* imaging. Orthotopic tumors were harvested for BMPR1A staining.

3. Fig 4A-C. the authors stated that “We found that BMP signaling was the only pathway that exhibited a significantly negative correlation with LAPT5 levels in both KIRC and KIRP patients” and focused on the LAPT5 and BMP pathway thereon. This reviewer could see that the reason the authors focused in the BMP pathway. However, it is equally possible that LAPT5 can promote lung-specific metastasis by regulating the pathways showing a positive correlation. Indeed, in Table S3,4, there are several pathways that show good positive correlation with LAPMT5. Therefore, the authors should provide a clear rationale for initially focusing on the BMP pathway in addition to negative correlation and bioactive BMPs in the lung.

Reply: Good question. As the reviewer mentioned, LAPT5 levels correlated with both positively and negatively regulated pathways in KIRC and KIRP cohorts. Among these, **ONE** negative pathway (BMP signaling) and 49 positive pathways were common in both KIRC and KIRP dataset. Because the majority of LAPT5 literature reported its negative regulation of membrane receptors [7-9], we decided to emphasize on signaling pathways that are suppressed by LAPT5 in this study. However, we could not rule out the involvement of positively correlated pathways in renal cancer lung metastasis. We have added these rationales to the results section and discussed it in the discussion section.

4. Authors have shown, in Fig3F, that KD or OE of LAMPT5 reduced and

increased sphere formation, respectively. According to Fig 4K, however, the BMPR1 protein levels are similar between control and LAMPT5-manipulated cells in the absence of exogenous BMP4. If LAMPT5-mediated BMPRA1 degradation is a key pathway for LAMPT5-induced sphere formation, how the results in Fig 3F can be explained in which presumably BMPR1 protein levels are similar between cells used in the experiment? The authors should collect spheres from these experiments and examined for the BMPR1 protein levels.

Reply: Good question. First, the experiment in Fig. 4K was repeated for several times and only one representative blot was shown. The original blot from another experiment is shown below. The quantification of these blots showed that BMPR1A level in Renca cells was downregulated when LAPT5 was overexpressed in the absence of exogenous BMP4, this is also consistent with the result obtained in 786-O cells from Figures 5M and 5N.

Second, per the reviewer's suggestion, we collected tumor spheres from repeated experiments in Figure 4G and examined BMPR1A protein levels. A decrease in BMPR1A expression was observed with LAPT5 overexpression and further enhanced by BMP4 treatment.

5. Fig 4G should be repeated with LAMPT5 KD-BMPR1A co-knockdown cells and/or LAMPT5 KD-DMH1 treatment.

Reply: We have performed additional experiments and added the data to new Figure 4H.

6. The authors have shown the lung metastasis-promoting roles of LAMPT5. However, this study completely lacks *in vivo* experiments supporting whether this is via LAMPT5KD-WWA-BMPR1A axis. The authors should perform *in vivo* experiments to clearly showing this.

Reply: We thank the reviewer for this suggestion. We have performed additional *in vivo* experiments and added the new data to Figure 6E. Briefly, we ectopically *Lap5* and/or *Bmpr1a* in Renca cells, and performed lung metastatic assays. These data (see below) further proved that LAMPT5 promoted lung met while BMPR1A inhibited lung met of Renca cells.

7. Fig5E-H, the authors performed several biochemical experiments to show that LAMPT5-WWA-BMPR1A forms a complex which leads to degradation of BMPR1A. However, a majority of experiments were performed with cells overexpressing LAMPT5-WWA-BMPR1A. Thus, similar experiments should be performed with endogenous proteins.

Reply: Good question. We have performed additional endogenous IP assay in parental 786O^{luc/eGFP} cells and the 786O^{LuM1a} derivative, these results (see below) confirmed the interactions between LAPTM5-WWP2 and WWP2-BMPR1A and are added to new Figure 5I.

8. Fig 5E-H, the authors should perform serial IP to clearly show LAMPT5-WWA- BMPR1 are indeed in the same complex.

Reply: We thank the reviewer for this suggestion. To address this question, we co-expressed Flag-LAPT5, HA-WWP2, and Myc-BMPR1A in 293T cells, we then performed the serial IP experiment using anti-Myc antibody for the first round and anti-HA antibody for the second round, and detected with anti-Flag antibody. We successfully detected the Flag-LAPT5 signal in the product from serial IP, which directly proved LAPT5-WWP2-BMPR1A were indeed in the same protein complex. These results are added as the new Figures 5E and 5F.

9. Fig 6, the experiments were performed with 293T cells only. The similar experiments should be performed in RCCs.

Reply: We thank the reviewer for this suggestion. We have repeated these experiments in 786-O cells have added the data as new Figures 6A-6D. The original 293T data are now moved to supplementary.

10. According to a previous paper “Functional Proteomics Mapping of a Human Signaling Pathway” (Genome Res, 2004), this paper showed that LAPMT5 interacts with Smurf2 and activated the TGF β pathway but not BMP pathway. The author should cite this paper and discuss the potential explanation for the differences between this paper and the current manuscript in the discussion. Also, the paper showed TGF β treatment increases LAPMT5 expression. This may provide a clue on why LAPMT5 is overexpressed in LM derivatives. The authors may want to test this in their experimental system.

Reply: Good question. Per the reviewer’s suggestions, we evaluated the effect of TGF- β in our system. First, we tested the effect of LAPTM5 on TGF- β signaling. We found that TGF- β treatment induced phosphorylation of Smad2/3 in both 786-O and Renca cells, but this was not affected by LAPTM5 manipulation (Figure S6D). Second, we tested the effect of TGF- β treatment on LAPTM5. We found a dose-dependent increase in LAPTM5

protein level in both 786-O and Renca cells after TGF- β treatment, suggesting that LAPTM5 may be upregulated in TGF- β -rich environment. This is consistent with the reviewer's predictions. We have added this to the discussion section of the revised manuscript.

However, interaction of LAPTM5 and SMURF2 was not detected in RCC cells (see below and new Figure S7D), and it could be the reason why we did not detect an inhibitory effect of LAPTM5 on TGF- β induced signaling in RCC cells.

Minor comments:

1. High resolution cell morphology S1c is needed.

Reply: We have added representative brightfield images of cells to this figure.

2. Does LAPMT4 KD revert the morphology of the RCC LM cell back to the par?

Reply: We did not observe an obvious morphology change in cells with LAPTM5 KD. We reasoned that morphological phenotype of the derivatives might not be directly related to the levels of LAPTM5.

3. It would be better to provide a bit more information on LAPTM5 on page 6 where it was first mentioned in the manuscript.

Reply: We thank the reviewer for pointing this out. We have added more text to introduce more background information on LAPTM5.

4. The figure numbers in the text should be according to the order of figures. (e.g pp7 Fig 3H and J come first than 3A.)

Reply: We have corrected this in the revised manuscript.

Reviewer #3:

This is a thorough and solid study introducing the role of LAPTM5 to sustain self-renewal and confer lung-tropism to RCC tumor cells. It was a pleasant experience to review this manuscript. Topic/hypothesis is of interest and significance. Logic flow is smooth and easy to follow. Experiment design and data presentation is clean and neat. Yet the reviewer founds several conceptual caveats that need to be addressed before this manuscript is accepted for publication in Nature Communication.

The authors tend to attribute RCC's lung tropism to the self-renewal program induced by LAPTM5. Reasoning that self-renewal/stemness per se is not an organ-specific phenotype, the authors' hypothesis may face more challenges and concerns, compared to other peer studies that focused on more organ-specific pathways.

Major comments:

1. One fundamental premise of the authors' hypothesis is the exceptional enrichment of BMPs in lung, which should suppress the activity of ESC transcription factors in all but the lung-tropic cells. However, the uneven expression of BMP in different organs is not well-evident in vivo. No data or reference were provided to support this notion except for a simple annotation at Line 192 [...bioactive BMPs (deficient in the bone and brain) from lung stroma...]. This is a major caveat.

Reply: We thank the reviewer for pointing this out and apologize for not stating the existing evidence clearly. In the original manuscript, we cited two papers (ref 13 and ref 38) that support the unusually high levels of BMP in the lung. In the first paper (Cell 2012), Gao et al. revealed that Coco induces dormant breast cancer cells to undergo reactivation by blocking lung-derived BMP ligands. Moreover, Coco induces a gene expression signature that is strongly associated with metastatic relapse to the lung, but not to bone and brain in patients. The authors studied the differences among organs in mouse model and showed that the bone and brain contain niches devoid of bioactive BMPs. These results supported the conclusion that BMPs served as the lung-specific antimetastatic signals [1]. In the second paper (Nat Commun. 2016), Song et al. found that GALNT14 promotes lung-specific metastasis of breast cancer cells by overcoming the inhibitory effect of lung-derived BMPs on self-renewal [2]. To provide more in vivo evidence, we performed IHC staining of p-smad 1/5/8 in various mouse organs, the results clearly showed the abundant phosphorylation of Smad 1/5/8 in lung stromal cells and partial phosphorylation in renal parenchymal cell, but none in the bone and brain (see below and Figure S6A). Together, these literature and data provided solid foundation for the hypothesis tested in this study.

2. Conceptually, “organotropism” needs to be interpreted from two aspects:
 a. Why only lung-derivatives can thrive in lung; b, Why the other organs are not favorable to lung-tropic cells? Question b is not discussed in this manuscript. Indeed, in most other peer studies focusing on organ-specific pathways, question b is usually self-evident (the indicated gene/pathway is not activated or even not present in other organs). However, this study is different from them. The authors are recommended to at least discuss why the self-renewal phenotype in lung-derivatives is less robust in brain and bone. Another relevant question is, can LAPT5 overexpression in bone- or brain- derivatives (not the parental cells) convert their organ tropism?

Reply: These are great questions. It is well established that microenvironment differs across organs and exerts distinct growth pressure on metastasis-initiating cells. In this paper, BMP signal suppresses metastasis in the lung while LAPT5 highly-expressed RCC cells could overcome this antimetastatic signal, and this is what we emphasized on (question a). We reasoned that in the bone and brain microenvironment, there must exist an organ-specific antimetastatic pressure such as plasminogen activator (PA) in the brain and Wnt family member 5a (Wnt5a) in the bone [10, 11]. Without a mechanism to overcome these antimetastatic signals, the high expression of LAPT5 could not allow lung-tropic RCC cells to thrive in the bone and brain (question b). To answer the last question, we ectopically expressed LAPT5 in bone- and brain-tropic RCC cells, and performed metastasis assays by tail vein injection of the cells, 21 days after implantation, lung metastases were analyzed by *in vivo* imaging. The results (see below) showed that ectopically expressed LAPT5 could also promote the lung tropism of bone and brain derivatives.

3. At least in some experiments (like sphere assay in Fig. 3F), the function of LAPT5 appears BMP-independent. The authors are recommended to clarify it.

Reply: This is a good question. We agree with the reviewer that in some of experiments the function of LAPT5 appears to be BMP-independent. To clarify this, we performed the sphere assays in Laptm5-silenced Renca^{LuM2b} cells in the absence and presence of DMH1, a BMPR inhibitor, the inhibitory effect of LAPT5 KD on sphere formation was largely reversed by DMH1 treatment (see below and Figure 4H), suggesting the function of LAPT5 is BMP-dependent. A possible explanation for the observed results is the involvement of endogenous BMPs in these experiments, i.e., an autocrine effect of self-generated BMP could not be ruled out.

4. It is a bit regretted that no pharmacological assay was performed for pre-clinical trial. Otherwise, the impact of this paper will be considerably increased.

Reply: We appreciated the reviewer's encouraging comments. We are currently carrying out pre-clinical therapeutic trials based on the findings in this work, which we consider to be beyond the scope of this manuscript and is more appropriate to publish separately in the future.

Minor comments:

1. Some “non-significant” results in Fig.3P are likely due to poor data performance (large deviation / outlier) instead of biological reason.

Reply: There was no Fig.3P in the original manuscript. If the reviewer is referring to Fig.2P, the photon flux varied dramatically and made the data look poor. We also performed H&E staining of the brain (data not show), which also showed similar results. However, the trend is pretty clear and we chose to report the data as it is.

2. Fig. 7H, LUNG-metastasis-free survival curve should be more relevant. Ideally, the LAMTP5 expression is expected to dictate ONLY lung-metastasis-free survival.

Reply: This is a good point. Since these data comes from the cohorts in TCGA, we could not get the information about the metastatic site of the RCC patients, therefore we could not plot the curve for ONLY lung-metastasis-free survival but could only plot for overall metastasis-free survival. In future work, we will report the lung-met-free survival when we have collected enough patient information from our own hospital.

REFERENCES

1. Gao, H., et al., *The BMP inhibitor Coco reactivates breast cancer cells at lung metastatic sites*. Cell, 2012. **150**(4): p. 764-79.
2. Song, K.H., et al., *GALNT14 promotes lung-specific breast cancer metastasis by modulating self-renewal and interaction with the lung microenvironment*. Nat Commun, 2016. **7**: p. 13796.
3. Png, K.J., et al., *MicroRNA-335 inhibits tumor reinitiation and is silenced through genetic and epigenetic mechanisms in human breast cancer*. Genes Dev, 2011. **25**(3): p. 226-31.
4. Hua, F., et al., *TRIB3 Interacts With beta-Catenin and TCF4 to Increase Stem Cell Features of Colorectal Cancer Stem Cells and Tumorigenesis*. Gastroenterology, 2019. **156**(3): p. 708-721 e15.
5. Yang, Y., et al., *E3 ligase WWP2 negatively regulates TLR3-mediated innate immune response by targeting TRIF for ubiquitination and degradation*. Proceedings of the National Academy of Sciences, 2013. **110**(13): p. 5115-5120.
6. Aki, D., et al., *The E3 ligases Itch and WWP2 cooperate to limit TH2 differentiation by enhancing signaling through the TCR*. Nat Immunol, 2018. **19**(7): p. 766-775.
7. Ouchida, R., et al., *A lysosomal protein negatively regulates surface T cell antigen receptor expression by promoting CD3zeta-chain degradation*. Immunity, 2008. **29**(1): p. 33-43.
8. Kawai, Y., et al., *LAPTM5 promotes lysosomal degradation of intracellular CD3 ζ but not of cell surface CD3 ζ* . Immunology and Cell Biology, 2014. **92**(6): p. 527-534.

9. Ouchida, R., T. Kurosaki, and J.Y. Wang, *A role for lysosomal-associated protein transmembrane 5 in the negative regulation of surface B cell receptor levels and B cell activation*. J Immunol, 2010. **185**(1): p. 294-301.
10. Valiente, M., et al., *Serpins promote cancer cell survival and vascular co-option in brain metastasis*. Cell, 2014. **156**(5): p. 1002-16.
11. Ren, D., et al., *Wnt5a induces and maintains prostate cancer cells dormancy in bone*. J Exp Med, 2019. **216**(2): p. 428-449.

REVIEWERS' COMMENTS

Reviewer #2 (Remarks to the Author):

The authors addressed all my comments and suggestions.

Reviewer #3 (Remarks to the Author):

The reviewer appreciated the authors' tremendous effort in successfully addressing most of the comments. Yet, the reviewer was not convinced by the response to the major comment #1. Given that the diverse abundance of BMP in different organs is important premise and foundation of this work, the reviewer strongly encourages the authors to further clarify this topic.

In the original manuscript, the authors stated that bioactive BMPs is abundant in lung stroma but "deficient in the bone and brain". After the first round of review, the reviewer argued that there is not sufficient evidence to support that BMP activity is exclusively high in lung. Thus, the authors' central hypothesis is questionable.

To address the reviewers' comment, the authors cited two papers (Gao et al, Cell 2012 and Song et al., Nat Commun. 2016) to support their original statement. Additionally, the authors provided IHC staining results to show the lack of BMP/Smad signaling in mouse bone and brain.

However, in the bone biology field, BMPs, or "Bone morphogenetic proteins" are considered to play important roles in bone development and bone turnover (Growth Factors. 2004;22(4):233-41; Front Mol Biosci. 2021, 5;8:593310). Indeed, the name of the BMP protein per se represents its presence in bone. Several bone metastasis studies also indicate that BMP signaling enhances bone metastasis of breast cancer cells through Smad pathway (Endocr Relat Cancer. 2017; 24(10): R349-R366.).

Overall, the authors are strongly encouraged to reconsider their premise and provide more thorough explanation/discussion to address the controversy between their observations and the well-accepted knowledge in bone biology.

Reviewer #4 (Remarks to the Author):

The authors have adequately and extensively replied to all of this reviewer's comments and concerns. As such, I recommend this article for publication.

Point-to-point response to reviewers:**NCOMMS-21-27982A****Lysosomal Protein Transmembrane 5 Promotes Lung-Specific Metastasis by Regulating BMPR1A Lysosomal Degradation****Reviewer #3:**

The reviewer appreciated the authors' tremendous effort in successfully addressing most of the comments. Yet, the reviewer was not convinced by the response to the major comment #1. Given that the diverse abundance of BMP in different organs is important premise and foundation of this work, the reviewer strongly encourages the authors to further clarify this topic.

In the original manuscript, the authors stated that bioactive BMPs is abundant in lung stroma but "deficient in the bone and brain". After the first round of review, the reviewer argued that there is not sufficient evidence to support that BMP activity is exclusively high in lung. Thus, the authors' central hypothesis is questionable.

To address the reviewers' comment, the authors cited two papers (Gao et al, Cell 2012 and Song et al., Nat Commun. 2016) to support their original statement. Additionally, the authors provided IHC staining results to show the lack of BMP/Smad signaling in mouse bone and brain.

However, in the bone biology field, BMPs, or "Bone morphogenetic proteins" are considered to play important roles in bone development and bone turnover (Growth Factors. 2004;22(4):233-41; Front Mol Biosci. 2021, 5;8:593310). Indeed, the name of the BMP protein per se represents its presence in bone. Several bone metastasis studies also indicate that BMP signaling enhances bone metastasis of breast cancer cells through Smad pathway (Endocr Relat Cancer. 2017; 24(10): R349–R366.).

Overall, the authors are strongly encouraged to reconsider their premise and provide more thorough explanation/discussion to address the controversy between their observations and the well-accepted knowledge in bone biology.

Reply: We thank the reviewer for pointing out this important concern about the contradictory role of BMP signaling in organ specific metastasis reported by us and other previous studies. We would like to explain/discuss it from the following perspectives.

First of all, BMPs are present in multiple organs but the expression levels are quite different. As the reviewer pointed out, BMPs do play critical roles in the development and maintenance of various tissues by regulating cell proliferation, differentiation and death, especially in the bone [1-3]. In accordance with this, nuclear accumulation of p-Smad 1/5/8 are detected in mouse chondrocytes from the growth plate, although not in most hematopoietic cells from the bone marrow [4] (supplementary Fig. 6a). Therefore, we agree with the reviewer that 'bioactive BMPs is abundant in lung stroma but deficient in the bone and brain' is an overstatement and had changed that in the revised

manuscript. Please note that, because BMP signaling is regulated by complex mechanisms [1, 5], BMP activity in organ environments is usually measured indirectly by detection of BMP-responsive p-Smad 1/5/8. Data from *Gao et al.* revealed that none of the solitary 4T07 and less than 6% of the solitary 4T1 metastatic breast cancer cells in the lung were p-Smad 1/5/8 negative, in contrast, 36.5% and 100% of the solitary 4T1 cells were p-Smad 1/5/8 negative in the bone marrow and brain parenchyma, respectively [4], which suggested the presence but relatively lower level of bioactive BMPs in the bone marrow than in the lung stroma. Data from *Song et al.* further supported this [6]. Hence, the distinct levels of BMP signaling could be one of the reasons behind the different functions of BMP observed in different organs.

Second, we reason that different organ might use different signaling molecules as their “primary weapon” to suppress cancer metastasis. Both our data and previous work in breast cancer [7] suggest that the lung uses BMP as the dominant signal to suppress the outgrowth of metastatic tumor cells, regardless of the tumor type. Thus, cell subpopulations that could overcome this signal (e.g., by overexpressing LAMPT5, Coco) could form metastasis in the lung. On the contrary, some other dominant signal (that are highly expressed in the metastatic tumor niche) might be used by the bone or the brain to suppress the metastatic colonization of circulating tumor cells, and only the cell clones that bear a certain trait and can overcome those signals could outgrow in the bone or the brain microenvironment. In this sense, the contradictory roles of BMP overserved in bone metastasis of various cancers (pro-metastatic in breast cancer [8] and prostate cancer [9]; anti-metastatic in breast cancer [10-13]) might be the combinatory effect of BMP manipulation and other factors in the bone microenvironment.

Thirdly, the distinct roles of BMP across different organs might also have something to do with the expression pattern of BMP subtypes. Although most BMP subtypes are uniformly expressed in multiple tissues during embryogenesis, the expression profile becomes tissue-specific after birth. For example, BMP-3, -4, -5 and -6 are highly expressed in the lung, whereas BMP-7 is abundantly expressed in the kidney in adult mice [14]; BMP-3 is abundantly expressed by osteoblasts and osteocytes in mouse skeletal system [15]. Whether different expression patterns of BMP subtypes contribute to their organ-specific function also require more studies.

Per the reviewer’s suggestion, we have also added a dedicated paragraph to the discussion section to address the controversy between our observations and the literature.

References

1. Katagiri, T. and T. Watabe, *Bone Morphogenetic Proteins*. Cold Spring Harbor Perspectives in Biology, 2016. **8**(6).
2. Chen, D., M. Zhao, and G.R. Mundy, *Bone morphogenetic proteins*. Growth Factors, 2004. **22**(4): p. 233–41.
3. Zou, M.L., et al., *The Smad Dependent TGF- β and BMP Signaling Pathway in Bone Remodeling and Therapies*. Front Mol Biosci, 2021. **8**: p. 593310.
4. Gao, H., et al., *The BMP inhibitor Coco reactivates breast cancer cells at lung metastatic sites*. Cell, 2012. **150**(4): p. 764–79.
5. Walsh, D.W., et al., *Extracellular BMP-antagonist regulation in development and disease: tied up in knots*. Trends Cell Biol, 2010. **20**(5): p. 244–56.
6. Song, K.H., et al., *GALNT14 promotes lung-specific breast cancer metastasis by modulating self-renewal and interaction with the lung microenvironment*. Nat Commun, 2016. **7**: p. 13796.
7. Martinez, J. and X.H.F. Zhang, *BMP/Coco antagonism as a deterministic factor of metastasis dormancy in lung*. Breast Cancer Research, 2013. **15**(1).
8. Zabkiewicz, C., et al., *Bone morphogenetic proteins, breast cancer, and bone metastases: striking the right balance*. Endocr Relat Cancer, 2017. **24**(10): p. R349–r366.
9. Nishimori, H., et al., *Prostate cancer cells and bone stromal cells mutually interact with each other through bone morphogenetic protein-mediated signals*. J Biol Chem, 2012. **287**(24): p. 20037–46.
10. Cao, Y., et al., *BMP4 Inhibits Breast Cancer Metastasis by Blocking Myeloid-Derived Suppressor Cell Activity*. Cancer Research, 2014. **74**(18): p. 5091–5102.
11. Ren, W., et al., *BMP9 inhibits the bone metastasis of breast cancer cells by downregulating CCN2 (connective tissue growth factor, CTGF) expression*. Mol Biol Rep, 2014. **41**(3): p. 1373–83.
12. Buijs, J.T., et al., *Bone morphogenetic protein 7 in the development and treatment of bone metastases from breast cancer*. Cancer Res, 2007. **67**(18): p. 8742–51.
13. Wang, K., et al., *BMP9 inhibits the proliferation and invasiveness of breast cancer cells MDA-MB-231*. J Cancer Res Clin Oncol, 2011. **137**(11): p. 1687–96.
14. Ozkaynak, E., et al., *Osteogenic protein-2. A new member of the transforming growth factor-beta superfamily expressed early in embryogenesis*. J Biol Chem, 1992. **267**(35): p. 25220–7.
15. Kokabu, S., et al., *BMP3 suppresses osteoblast differentiation of bone marrow stromal cells via interaction with Acvr2b*. Mol Endocrinol, 2012. **26**(1): p. 87–94.